



# An inter-laboratory comparison of aerosol inorganic ion measurements by Ion Chromatography: implications for aerosol pH estimate

Jingsha Xu[1], Shaojie Song[2], Roy M. Harrison[1], Congbo Song[1], Lianfang Wei[3], Qiang Zhang[4], Yele Sun[3], Lu Lei[3], Chao Zhang[5], Xiaohong Yao[5, 6], Dihui Chen[5], Weijun Li[7], Miaomiao Wu[7], Hezhong Tian[8], Lining Luo[8], Shengrui Tong[9], Weiran Li[9], Junling Wang[10], Guoliang Shi[11], Yanqi Huangfu[11], Yingze Tian[11], Baozhu Ge[3], Shaoli Su[12], Chao Peng[12], Yang Chen[12], Fumo Yang[13], Aleksandra Mihajlidi-Zelić[14], Dragana Đorđević[14], Stefan J. Swift[15], Imogen Andrews[15], Jacqueline F. Hamilton[15], Ye Sun[16], Agung Kramawijaya[1], Jinxiu Han[1], Supattarachai Saksakulkrai[1], Clarissa Baldo[1], Siqi Hou[1], Feixue Zheng[17], Kaspar R. Daellenbach[17], Chao Yan[17], Yongchun Liu[17], Markku Kulmala[17], Pingqing Fu[4], Zongbo Shi*[1]

School of Geography Earth and Environmental Science, University of Birmingham, Birmingham, B15 2TT, UK
School of Engineering and Applied Sciences, Harvard University, Cambridge, MA 02138, USA
State Key Laboratory of Atmospheric Boundary Layer Physics and Atmospheric Chemistry, Institute of Atmospheric Physics, Chinese Academy of Sciences, Beijing, 100029, China
Institute of Surface-Earth System Science, Tianjin University, Tianjin, 300072, China
Frontiers Science Center for Deep Ocean Multispheres and Earth System, and Key Laboratory of Marine Environment and Ecology, Ministry of Education of China, Ocean University of China, Qingdao 266100, China
Laboratory for Marine Ecology and Environmental Sciences, Qingdao National Laboratory for Marine Science and Technology, Qingdao 266071, China
Department of Atmospheric Sciences, School of Earth Sciences, Zhejiang University, Hangzhou, 310027, China
Center for Atmospheric Environmental Studies, Beijing Normal University, Beijing, 100875, China
State Key Laboratory of Structural Chemistry of Unstable and Stable Species, Institute of Chemistry, Chinese Academy of Sciences, Beijing, 100190, China
School of Environment, Tsinghua University, Beijing, 100084, China
State Environmental Protection Key Laboratory of Urban Ambient Air Particulate Matter Pollution Prevention and Control, Center for Urban Transport Emission Research, College of Environmental Science and Engineering, Nankai University, Tianjin, 300350, China
Research Center for Atmospheric Environment, Chongqing Institute of Green and Intelligent Technology, Chinese Academy of Sciences, Chongqing, 400714, China
Department of Environmental Science and Engineering, Sichuan University, Chengdu, 610065, China
Centre of Excellence in Environmental Chemistry and Engineering – ICTM, University of Belgrade, Njegoševa 12 (Studentski trg 14–16), Belgrade, Serbia
Department of Chemistry, University of York, York, YO10 5DD, UK
School of Space and Environment, Beihang University, Beijing, 100191, China
Beijing Advanced Innovation Center for Soft Matter Science and Engineering, Beijing University of Chemical Technology, Beijing, 100029, China

Correspondence: Zongbo Shi (Z.Shi@bham.ac.uk)





**ABSTRACT**
Water soluble inorganic ions such as ammonium, nitrate, and sulfate are major components of fine
aerosols in the atmosphere and are widely used in the estimation of aerosol acidity. However, different
experimental practices and instrumentation may lead to uncertainties in ion concentrations. Here, an
inter-comparison experiment was conducted in 10 different laboratories (labs) to investigate the
consistency of inorganic ion concentrations and resultant aerosol acidity estimates using the same set
of aerosol filter samples. The results mostly exhibited good agreement for major ions $Cl^-$, $SO_4^{2-}$, $NO_3^-$,
$NH_4^+$ and $K^+$. However, $F^-$, $Mg^{2+}$ and $Ca^{2+}$ were observed with more variations across the different
labs. The Aerosol Chemical Speciation Monitor (ACSM) data of non-refractory $SO_4^{2-}$, $NO_3^-$, $NH_4^+$
generally correlated very well with the filter analysis based data in our study, but the absolute
concentrations differ by up to 42%. $Cl^-$ from the two methods are correlated but the concentration
differ by more than 3 times. The analyses of certified reference materials (CRMs) generally showed
good recovery of all ions in all the labs, the majority of which ranged between 90% and 110%. Better
agreements were found for $Cl^-$, $SO_4^{2-}$, $NO_3^-$, $NH_4^+$ and $K^+$ across the labs after their concentrations
were corrected with CRM recoveries; the coefficient of variation (CV) of $Cl^-$, $SO_4^{2-}$, $NO_3^-$, $NH_4^+$ and
$K^+$ decreased 1.7%, 3.4%, 3.4%, 1.2% and 2.6%, respectively, after CRM correction. We found that
the ratio of anion to cation equivalent concentrations (AE/CE) is not a good indicator for aerosol
acidity estimates, as the results in different labs did not agree well with each other. Ion balance (anions
− cations) calculated from $SO_4^{2-}$, $NO_3^-$ and $NH_4^+$ gave more consistent results, because of their
relatively large concentrations and good agreement among different labs. In situ aerosol pH calculated
from the ISORROPIA-II thermodynamic equilibrium model with measured ion and ammonia
concentrations showed a similar trend and good agreement across the 10 labs. Our results indicate
that although there are important uncertainties in aerosol ion concentration measurements, the
estimated aerosol pH from the ISORROPIA-II model is more consistent.
**Keywords:** $PM_{2.5}$, inorganic ions, aerosol acidity, ion balance, thermodynamic model



## 1.    INTRODUCTION

Water-soluble inorganic ions (WSII), consisting of $F^-$, $Cl^-$, $NO_2^-$, $NO_3^-$, $SO_4^{2-}$, $NH_4^+$, $Na^+$, $K^+$, $Mg^{2+}$ and $Ca^{2+}$, are a major component of atmospheric aerosols and can contribute up to 77% of $PM_{2.5}$ (particulate matter with aerodynamic diameter $\leq 2.5$ μm) mass (Xu et al., 2019a). Secondary inorganic aerosols (SIA) including sulfate, nitrate and ammonium (SNA) often dominate water-soluble ionic species in $PM_{2.5}$, and were reported to account for more than 90% of WSII in Sichuan, China (Tian et al., 2017). In Beijing, the average SNA concentrations can range from $4.2 \pm 2.9$ μg/m$^3$ in non-haze days to $85.9 \pm 22.4$ μg/m$^3$ in heavily polluted days, and contribute to 15%-49% of $PM_{2.5}$ (Li et al., 2016). SNA can greatly influence air pollution, visibility, aerosol acidity and hygroscopicity, which are driving factors affecting aerosol-phase pH and chemistry and the uptake of gaseous species by particles (Shon et al., 2012; Xue et al., 2011; Zhang et al., 2019). Hence, the study of WSII is of great interest due to their adverse impacts.

WSII in aerosols were reported to be analyzed by multiple techniques such as $Cl^-$ by spectrophotometry, and $Ca^{2+}$ and $Mg^{2+}$ by flame atomic absorption in the early 1980s (Harrison and Pio, 1983). However, this was very time-consuming as different ions required to be analyzed by different techniques. Ion chromatography (IC), which was first introduced in 1975 (Buchberger, 2001), was applied in many studies for routine measurement of atmospheric WSII due to its fast, accurate and sensitive determination in a single run (Heckenberg and Haddad, 1984; Baltensperger and Hertz, 1985). IC can be coupled with diverse detection techniques for ion analysis, such as suppressed conductivity, UV-VIS absorbance, amperometry, potentiometry, mass spectrometry, etc. (Buchberger, 2001). It has been used in various atmospheric studies for many years and is still widely applied nowadays, such as in the investigation of WSII in size-segregated aerosols (Li et al., 2013; Zhao et al., 2011; Đorđević et al., 2012), fine aerosols (Fan et al., 2017; He et al., 2017; Liu et al., 2017a) and coarse aerosols (Li et al., 2014; Guo et al., 2011; Mkoma et al., 2009). IC can also be used



for the determination of both water soluble organic and inorganic ions (Yu et al., 2004; Karthikeyan
and Balasubramanian, 2006).
Aerosol ion concentrations can also be measured by online methods such as the Aerosol Chemical
Speciation Monitor (ACSM) or Aerosol Mass Spectrometer (AMS) (Ng et al., 2011; Sun et al., 2012).
During the recent Atmospheric Pollution and Human Health in a Chinese Megacity (APHH-China)
campaigns (Shi et al., 2019), we observed important discrepancies between offline aerosol IC
observations from different labs and between online AMS and offline IC methods. This prompted us
to carry out this intercomparison exercise.
The IC method had been validated by a common reference standard - NIST SRM 1648 (urban
particulate matter) and the results for Na, K, S and $NH_4^+$ were compared with those from other
suitable alternative analytical techniques such as AAS, UV-VIS and PIXE in previous studies
(Karthikeyan and Balasubramanian, 2006). However, to the best of our knowledge, no investigation
has been conducted to compare the results of different laboratories (labs) for such an important and
widely used simple technique.
The aim of this work is to 1) examine the consistency of ion concentrations measured by various labs
and by ACSM, 2) explore the impact of the inter-lab variability in ion concentration measurements
on aerosol acidity estimates, and 3) provide recommendations for improving future WSII analysis by
IC.

**2.    EXPERIMENTAL**
**2.1    Participating Laboratories**
Ten laboratories from China, United Kingdom and Serbia were invited to take part in the inter-
laboratory comparison of atmospheric inorganic ions, which are listed as follows: University of
Birmingham; University of York; University of Belgrade; Zhejiang University; Nankai University;
Ocean University of China; Beijing Normal University; Chongqing Institute of Green and Intelligent





Technology, Chinese Academy of Sciences; Institute of Chemistry, Chinese Academy of Sciences;
Institute of Atmospheric Physics, Chinese Academy of Sciences. The participating laboratories were
randomly coded from Lab-1 to Lab-10 and not related to the above order.

**2.2      Sample and Data Collection**
Eight daily $PM_{2.5}$ samples were collected on quartz filters (total area: 406.5cm$^2$) from 16$^{th}$-23$^{rd}$
January 2019 by a high-volume air sampler (1.13 m$^3$ min$^{-1}$; Tisch Environmental Inc., USA) at an
urban site, located at the Institute of Atmospheric Physics (IAP) of the Chinese Academy of Sciences
in Beijing, China. The sampling site (116.39E, 39.98N) is located between the North Third Ring Road
and North Fourth Ring Road, and approximately 200 m from the G6 Highway. It is 8 m above the
ground and surrounded by high-density roads and buildings; detailed information regarding the
sampling site can be found elsewhere (Shi et al., 2019). Apart from the aerosol samples, 5 field blank
filters were also collected in the same manner with the pump off. All ion concentrations in this study
were corrected by the values obtained from field blanks. Hourly $PM_{2.5}$ mass concentrations were
obtained from a nearby Olympic Park station, the China National Environmental Monitoring Network
(CNEM) website. Shi et al. (2019) showed that the $PM_{2.5}$ data at this station are close to those
observed at IAP during the APHH-China campaigns. The close observed $PM_{2.5}$ concentrations at
different air quality stations in Beijing provide further reassurance of the representability of the
observed concentration at Olympic Park. The original hourly data was averaged to 24 h for better
comparison.

An Aerodyne Time-of-Flight Aerosol Chemical Speciation Monitor (ToF-ACSM) with a $PM_{2.5}$
aerodynamic lens was also deployed on the same roof of the building at IAP for real-time
measurements of non-refractory (NR) chemical species (Organics, Cl$^-$, NO$_3^-$, SO$_4^{2-}$ and NH$_4^+$) in
$PM_{2.5}$ (NR-$PM_{2.5}$) with 2 min time resolution (Sun et al., 2020). Another ToF-ACSM was also used





to measure the $PM_{2.5}$-associated non-refractory chemical species at the Beijing University of
Chemical Technology (BUCT), which is located at the west third-Ring Road of Beijing and
approximately 10 km away from the sampling location of IAP. The concentrations of non-refractory
species were calculated from mass spectra using a fragmentation table (Allan et al., 2004). The ToF-
ACSM data were then averaged to 24h for a comparison with those from filter analysis in our study.
Note that the ToF-ACSM data at IAP on 19[th] and 20[th] and data at BUCT on 17[th] and 18[th] are excluded
from the comparison due to the maintenance of the instrument. An ammonia analyzer (DLT‐100,
Los Gatos Research LGR, USA) which applies a unique laser absorption technology called off-axis
integrated cavity output spectroscopy was used for the ambient $NH_3$ measurements. It has a precision
of 0.2 ppb and the original data with 5 min intervals were averaged to 24 h for the calculation of
aerosol pH.    More information on $NH_3$ measurement can be found elsewhere (Ge et al., 2019).

**2.3      Sample Analysis**
Filter cuts of $5cm^2$ and $6cm^2$ from the same set of samples were used for extraction in 10 labs. Filters
were extracted ultrasonically for 30 minutes with 10 ml ultrapure water in all laboratories and then
filtered before IC analysis. The instrument details are given in Table 1. In total, 9 ionic species were
reported: $F^-$, $Cl^-$, $SO_4^{2-}$, $NO_3^-$, $Na^+$, $NH_4^+$, $K^+$, $Mg^{2+}$ and $Ca^{2+}$. Other ions including $Br^-$, $NO_2^-$, $PO_4^{3-}$
and $Li^+$ were not included due to their relatively low concentrations in aerosol samples.

Certified reference materials (CRM) were also determined for quality control. CRM for cations
(CRM-C, Multi Cation Standard 1 for IC, Sigma-Aldrich) contains 200mg/L $Na^+$, 200mg/L $K^+$,
50mg/L $Li^+$, 200mg/L $Mg^{2+}$, 1000mg/L $Ca^{2+}$ and 400mg/L $NH_4^+$. CRM for anion (CRM-A, Multi
Anion Standard 1 for IC, Sigma-Aldrich) contains 3mg/L $F^-$, 10mg/L $Cl^-$, 20mg/L $Br^-$, 20mg/L $NO_3^-$,
20mg/L $SO_4^{2-}$ and 30mg/L $PO_4^{3-}$. CRM-C and CRM-A were diluted 180 and 6 times, respectively.



20mL of the diluted CRM solutions were marked as unknown solutions and sent along with the
aerosol samples to each lab for analysis. All CRM solutions were measured by each lab as unknown
samples. All filters and solutions were kept frozen during transportation to prevent any loss due to
volatilization.
**Table 1.** Summary of instrument and method details in 10 laboratories.

| Lab No. | Instrument model (Ion Chromatograph) | | Columns & suppressor | | Eluent | |
|---|---|---|---|---|---|---|
| | Anions | Cations | Anions | Cations | Anions | Cations |
| 1 | Dionex AQUNION-1100 | Dionex AQUNION-1100 | IonPac™ AS11-HC separation column; IonPac™ AG11-HC guard column; suppressor ASRS 300 | IonPac™ CS12A separation column; IonPac™ CG12A guard column; suppressor CSRS 300; | 30 mM KOH; 1.0 ml/min. | 20 mM methansulfonic acid; 1.0 ml/min. |
| 2 | Dionex ICS-1100 | Dionex ICS-1100 | IonPac™ AS11-HC separation column; IonPac™ AG11-HC guard column; suppressor ASRS 500 | IonPac™ CS12A separation column; IonPac™ CG12A guard column; suppressor CSRS 500 | KOH with gradient variation from 0 to 30 mM; 0.38 ml/min. | 15 mM methansulfonic acid; 0.25 ml/min |
| 3 | Dionex ICS-600 | Dionex ICS-600 | IonPac™ AS11-HC separation column; IonPac™ AG11-HC guard column; suppressor ASRS 300 | IonPac™ CS12A separation column; IonPac™ CG12A guard column; suppressor CSRS 300 | 20 mM KOH; 1.0 ml/min. | 20 mM methansulfonic acid; 1.0 ml/min |
| 4 | Dionex 600 | Dionex ICS 2100 | IonPac™ AS11 separation column; IonPac™ AG11 guard column; suppressor ASRS 300 | IonPac™ CS12A separation column; IonPac™ CG12A guard column; suppressor CSRS 300 | 30 mM KOH; 1.0 ml/min | 20 mM methansulfonic acid; 1.0 ml/min |
| 5 | Ion Chromotragraph (ECO) | Ion Chromotragraph (ECO) | Metrosep A5-150 separation column; Metrosep A SUPP 4/5 Guard/4.0 guard column; suppressor MSM | Metrosep C4-150 separation column | 3.2 mM Na2CO3-1.0mM NaHCO3; 0.7 ml/min | 1.7 mM nitric acid - 0.7mM dipicolinic acid; 0.9 ml/min |
| 6 | Metrohm (940 Professional IC Vario) | Metrohm (940 Professional IC Vario) | Metrohm A SUPP 5-250 separation column; Metrohm A SUPP 10-250 guard column; suppressor MSM-A Rotor | METROSEP C6-150 separation column; Metrohm C4 guard column | 3.2 mM Na2CO3-1.0mM NaHCO3; 0.7 ml/min | 1.7 mM nitric acid - 1.7mM dipicolinic acid; 0.9 ml/min |
| 7 | Dionex ICS600 | Dionex ICS600 | IonPac™ AS11-HC separation column; IonPac™ AG11-HC guard column; suppressor ASRS | IonPac™ CS12A separation column; IonPac™ CG12A guard column; suppressor CSRS | 30 mM KOH; 1ml/min | 20 mM methansulfonic acid; 1.0 ml/min |
| 8 | Dionex ICS-900 | Dionex ICS-900 | IonPac™ AS14 separation column; IonPac™ AG14 guard column; suppressor Dionex CCRS 500 | IonPac™ CS12A separation column; IonPac™ CG12A guard column; suppressor Dionex CCRS 500 | 3.5 mM Na2CO3-1.0mM NaHCO3; 1.2 ml/min | 20 mM methansulfonic acid; 1.0 ml/min |
| 9 | Dionex ICS-1100 | Dionex ICS-1100 | IonPac™ RFIC™ AS14A separation column; IonPac™ RFIC™ AG14A Guard column | IonPac™ RFIC™ CS12A separation column; IonPac™ RFIC™ CG12A Guard column | 8.0 mM Na2CO3-1.0mM NaHCO3; 1.0 ml/min | 20 mM methansulfonic acid; 1.0 ml/min |
| 10 | Dionex ICS-2100 | Dionex INTEGRION HPIC | IonPac™ AS15 separation column; IonPac™ AG15 guard column; suppressor ADRS 600 | IonPac™ CS12A separation column; IonPac™ CG12A guard column; suppressor CERS 500; | 38mM KOH; 0.3 ml/min. | 20 mM methansulfonic acid; 1.0 ml/min. |






**2.4      Coefficient of Divergence Analysis**
In order to investigate the differences of ionic concentrations measured by different labs, the
Pearson's correlation coefficient (R) and the coefficient of divergence (COD) were applied.
COD is a parameter to evaluate the degree of uniformity or divergence of two datasets. COD
and R were computed for $Lab_j$/Lab-Median pairs, of which $Lab_j$ indicates the results of each
lab and Lab-Median represents the median values of 10 labs. Median values are chosen here to
better represent the theoretical true concentrations of the ions, as there are some outliers in
some labs, and the averages may be less representative. The results of COD and R were also
computed for $Lab_j$/Lab-Mean, $Lab_j$/Lab-Upper and $Lab_j$/Lab-Lower pairs (Supplemental
Information Fig. S1-S3), where Lab-Mean, Lab-Upper and Lab-Lower represent the mean
value, upper values (84% percentile) and lower values (16% percentile) of ion concentrations
measured by 10 labs. COD of ionic concentrations of two datasets is determined as follows:
$$COD_{jk} = \sqrt{\frac{1}{P}\sum_{i=1}^{P}(\frac{X_{ij}-X_{ik}}{X_{ij}+X_{ik}})^2} \qquad (1)$$
where j represents the ion concentrations measured by an individual lab-j, k stands for the
median ion concentrations of 10 labs, P is the number of samples. $X_{ij}$ and $X_{ik}$ represent the
concentration of ion i measured by lab-j and the median concentration of ion i measured by 10
labs, respectively. COD value equals to 0 implies no difference between two datasets, while a
COD of 1 means absolute heterogeneity and maximum difference between two datasets (Liu
et al., 2017c). A COD value of 0.2 is applied as an indicator for similarity and variability
(Krudysz et al., 2008). A higher COD (>0.2) implies variability between two datasets, while
lower COD (<0.2) indicates similarity between them. A COD value of 0.269 is used here as an
indicator as well, as this value was also applied in other studies (Kamal et al., 2016;
Wongphatarakul et al., 1998). Overall, lower COD (<0.2) and higher R (>0.8) of the lab suggest





the similar variation pattern and similar ion concentrations of this lab with the median values
of 10 labs.

**2.5      ISORROPIA-II**
ISORROPIA-II is a thermodynamic equilibrium model for predicting the composition and
physical state of atmospheric inorganic aerosols (available at http://isorropia.eas.gatech.edu)
(Fountoukis and Nenes, 2007). It was applied in this study to calculate the aerosol water content
(AWC) and pH. Aerosol pH in this study ($pH_i$) was defined as the molality-based hydrogen ion
activity on a logarithmic scale, calculated applying the following equation (Jia et al., 2018;
Song et al., 2019):
$$pH_i = -\log_{10}\left(a_{H^+_{(aq)}}\right) = -\log_{10}\left(m_{H^+_{(aq)}}\gamma_{H^+_{(aq)}}/m^\Theta\right) \tag{2}$$
where $a_{H^+_{(aq)}}$ represents hydrogen ion activity in aqueous solution, $H^+_{(aq)}$. $m_{H^+_{(aq)}}$ and $\gamma_{H^+_{(aq)}}$
represent the molality and the molality-based activity coefficient of $H^+_{(aq)}$, respectively. $m^\Theta$
is the standard molality (1 mol $kg^{-1}$). Model inputs include aerosol-phase $Cl^-$, $SO_4^{2-}$, $NO_3^-$, $Na^+$,
$NH_4^+$, $K^+$, $Mg^{2+}$, $Ca^{2+}$ and gas-phase $NH_3$ concentrations, along with daily averaged
temperature and relative humidity (Table S2). In this study, the model was run only in forward
mode (gas and aerosol concentrations of species are fixed) in the thermodynamically
metastable phase state, assuming salts do not precipitate under supersaturated conditions. More
information regarding applications of ISORROPIA-II can be found in other studies (Guo et al.,
2016; Weber et al., 2016; Song et al., 2018).



## 3. RESULTS AND DISCUSSION

### 3.1 Quality Assurance and Quality Control (QA & QC)

#### 3.1.1 Certified reference materials (CRM) - Recovery and repeatability

Certified reference materials for both cations and anions were investigated for quality control.

CRM-C and CRM-A were analyzed three consecutive times in each lab. The recovery of each

ion was determined as the ratio of measured concentration divided by its certified concentration

in percentage. The results of recovery of all ions are listed in Table 2.

**Table 2.** Recovery (%) of water-soluble inorganic ions in certified reference materials measured by 10 laboratories.

| Lab NO. | $F^-$ | $Cl^-$ | $SO_4^{2-}$ | $NO_3^-$ | $Na^+$ | $NH_4^+$ | $K^+$ | $Mg^{2+}$ | $Ca^{2+}$ |
|---------|-------|--------|-------------|----------|--------|----------|-------|-----------|-----------|
| 1 | $111.8 \pm 0.2$ | $107.6 \pm 0.1$ | $108.5 \pm 2.4$ | $110 \pm 0.5$ | $98.2 \pm 0.0$ | $108.7 \pm 0.3$ | $99.4 \pm 0.2$ | $95.6 \pm 0.3$ | $99.6 \pm 0.6$ |
| 2 | $89.1 \pm 0.4$ | $95.1 \pm 0.2$ | $94.0 \pm 1.0$ | $94.5 \pm 0.5$ | $102.2 \pm 1.0$ | $135.0 \pm 6.0$ | $94.9 \pm 4.6$ | $95.9 \pm 0.2$ | $92.8 \pm 0.5$ |
| 3 | $101 \pm 1.4$ | $95.9 \pm 0.3$ | $132.4 \pm 31.4$ | $97.1 \pm 1.0$ | $91.4 \pm 0.1$ | $93.5 \pm 0.2$ | $92.4 \pm 0.2$ | $105.5 \pm 0.3$ | $98.7 \pm 0.4$ |
| 4 | $94.1 \pm 4.0$ | $90.4 \pm 0.2$ | $91.9 \pm 1.2$ | $91.7 \pm 1.4$ | $93.3 \pm 1.7$ | $112.2 \pm 0.6$ | $92.0 \pm 2.8$ | $98.9 \pm 2.0$ | $100.4 \pm 1.1$ |
| 5 | $94.0 \pm 3.1$ | $99.0 \pm 0.0$ | $92.4 \pm 0.9$ | $97.7 \pm 0.0$ | $85.9 \pm 3.2$ | $89.3 \pm 0.5$ | $92.1 \pm 4.9$ | $96.1 \pm 0.6$ | $101.7 \pm 3.0$ |
| 6 | $93.3 \pm 0.3$ | $110.8 \pm 0.5$ | $89.2 \pm 0.1$ | $91.4 \pm 0.2$ | $98.2 \pm 1.1$ | $88.4 \pm 1.1$ | $92.2 \pm 4.9$ | $102.0 \pm 2.1$ | $102.6 \pm 1.2$ |
| 7 | $89.4 \pm 2.7$ | $114.5 \pm 21.3$ | $100.8 \pm 0.0$ | $105.2 \pm 0.2$ | $97.0 \pm 1.3$ | $107.5 \pm 0.8$ | $72.1 \pm 0.8$ | $93.5 \pm 0.4$ | $91.9 \pm 1.1$ |
| 8 | $92.0 \pm 0.0$ | $96.6 \pm 0.7$ | $97.4 \pm 1.1$ | $96.2 \pm 1.2$ | $97.3 \pm 0.0$ | $93.8 \pm 0.3$ | $97.3 \pm 0.9$ | $94.0 \pm 2.1$ | $89.3 \pm 0.6$ |
| 9 | $102.6 \pm 1.5$ | $105.9 \pm 1.0$ | $101.9 \pm 4.5$ | $99.1 \pm 3.5$ | $101.2 \pm 0.1$ | $110.6 \pm 0.2$ | $103.0 \pm 0.0$ | $99.7 \pm 0.2$ | $102.2 \pm 0.3$ |
| 10 | $103.4 \pm 1.6$ | $103.5 \pm 0.7$ | $99.0 \pm 9.3$ | $114.2 \pm 2.5$ | $95.3 \pm 4.1$ | $91.0 \pm 4.1$ | $91.5 \pm 4.7$ | $94.8 \pm 3.8$ | $96.3 \pm 2.1$ |

As reported in Table 2, most ions were observed with a recovery in the range 90% - 110%

among 10 laboratories. However, $SO_4^{2-}$ in Lab-3 and $NH_4^+$ in Lab-2 were overestimated, which

resulted in the recovery of 132.4%±31.4% and 135.0%±6.0%, respectively. The standard

deviation of $SO_4^{2-}$ measured by Lab-3 was the largest (31.4%), followed by $Cl^-$ measured by

Lab-7 (21.3%), which indicated their poor repeatability. Even though $NH_4^+$ in Lab-2 was

observed with high recovery, its deviation of three repeats was relatively small, which may be

attributable to the evaporation of ammonium in calibration standards in Lab-2; hence, the level

it represented was higher than its real concentration. $K^+$ in Lab-7 was underestimated, and was





observed with a recovery of only 72.1%±0.8%. This may be due to contamination in the water
blanks or the IC system.

**3.1.2    Detection limits**
The detection limits (DLs) in this study were calculated as:
$DL = 3 \times SD_i$                                                                                        (3)
where $SD_i$ is the standard deviation of the blank filters. The mean concentrations of the ions in
blanks and DLs (3SD) of all ions are provided in Table 3.
**Table 3.** Mean filter blank concentrations and detection limits (3SD) (ng/m$^3$) of ions measured by 10
laboratories.

| Lab | F$^-$ mean | F$^-$ 3SD | Cl$^-$ mean | Cl$^-$ 3SD | SO$_4^{2-}$ mean | SO$_4^{2-}$ 3SD | NO$_3^-$ mean | NO$_3^-$ 3SD | Na$^+$ mean | Na$^+$ 3SD | NH$_4^+$ mean | NH$_4^+$ 3SD | K$^+$ mean | K$^+$ 3SD | Mg$^{2+}$ mean | Mg$^{2+}$ 3SD | Ca$^{2+}$ mean | Ca$^{2+}$ 3SD |
|---|---|---|---|---|---|---|---|---|---|---|---|---|---|---|---|---|---|---|
| 1 | 2.3 | 4.0 | 33.2 | 31.5 | 74.2 | 12.7 | 64.2 | 7.1 | 78.3 | 31.3 | 37.2 | 16.6 | 7.9 | 19.6. | 3.4 | 3.9. | 50.0 | 18.2. |
| 2 | 0.2 | 0.4 | 10.9 | 11.3 | 15.6 | 2.5 | 35.3 | 14.7 | 11.5 | 8.0 | 20.8 | 5.0 | 3.4 | 1.2 | 3.2 | 6.8 | 38.1 | 54.6 |
| 3 | 2.8 | 2.0 | 6.3 | 2.7 | 8.7 | 11.7 | 15.3 | 10.5 | 0.5 | 3.4 | 9.6 | 3.9 | 0.0 | 0.0 | 0.0 | 0.0 | 6.8 | 18.8 |
| 4 | 59.6 | 195.2 | 103.6 | 229.3 | 85.3 | 25.9 | 50.3 | 159.6 | 22.8 | 29.4 | 59.6 | 123.2 | 19.1 | 26.5 | 10.1 | 1.9 | 376.4 | 90.4 |
| 5 | 4.2 | 2.7 | 50.9 | 98.6 | 33.4 | 40.8 | 25.7 | 116.5 | 51.6. | 57.7. | 46.3 | 54.7 | 22.6 | 9.4 | 45.4 | 9.4 | 268.2 | 49.8 |
| 6 | n.a. | n.a. | 251.6 | 7.4 | 55.1 | 53.6 | 24.5 | 0.0 | 56.6 | 35.9 | 35.0 | 46.1 | n.a. | n.a. | n.a. | n.a. | n.a. | n.a. |
| 7 | 2.1 | 3.9 | 11.4 | 11.8 | 37.9 | 32.7 | 14.5 | 40.3 | 6.1 | 0.0 | 5.8 | 0.0 | 4.9 | 12.4 | 1.4 | 4.5 | 7.6 | 17.3 |
| 8 | n.a. | n.a. | 29.0 | 32.8 | 17.4 | 26.1 | n.a. | n.a. | 8.7 | 22.6 | n.a. | n.a. | n.a. | n.a. | n.a. | n.a. | 20.3 | 27.1 |
| 9 | n.a. | n.a. | n.a. | n.a. | 34.8 | 32.8 | 39.5 | 22.7 | 47.4 | 12.1 | 21.2 | 8.3 | 10.1 | 5.4 | 1.6 | 0.2 | 7.1 | 32.4 |
| 10 | 29.4 | 1.3 | 19.5 | 21.1 | 59.4 | 21.3 | 78.8 | 102.2 | 24.6 | 53.3 | 33.5 | 34.2 | 31.3 | 82.1 | 4.3 | 0.0 | 10.2 | 14.6 |

*Note: The detection limits were calculated based on large-volume sampling (total filter size: 406.5 cm$^2$; total*
*sampling volume: 1560 m$^3$); n.a.: not available due to no relevant peaks being identified in the chromatography.*





### 3.2 Mass Concentrations of PM₂.₅ and Inorganic Ions

#### 3.2.1 PM₂.₅ and ion concentrations

The results for PM$_{2.5}$ and all inorganic ion concentrations measured by 10 labs are presented in Fig. 1. During January 16$^{th}$ – 23$^{rd}$ 2019, the daily mean PM$_{2.5}$ ranged from 8.4 to 53.8 µg/m$^3$, with an average of 31.4 µg/m$^3$. Among them, January 16$^{th}$, 17$^{th}$ and 18$^{th}$ were deemed moderately polluted days with PM$_{2.5}$ concentration > 35 µg/m$^3$, while the rest were non-haze days with PM$_{2.5}$ concentrations falling in the range of 8.4-27.9 µg/m$^3$.




**Fig. 1.** The time series of mass concentrations of PM$_{2.5}$ and ions





The time series of all inorganic ions are also shown in Fig. 1 to demonstrate the consistency
among different laboratories. In Fig. 1, $Cl^-$, $NO_3^-$, $SO_4^{2-}$ and $NH_4^+$ showed a similar trend to
$PM_{2.5}$ and good correlations among the 10 labs, suggesting the consistency and reliability of
using Ion Chromatography for analysing these ions, despite various instruments and analysing
methods. Larger variations of $Cl^-$, $NO_3^-$, $SO_4^{2-}$ and $NH_4^+$ concentrations between different
laboratories were observed in moderately polluted days, whereas results for the non-haze days,
especially for 19[th] and 20[th], were observed with good agreement in 10 labs.

The average SNA concentrations of 8 samples varied from 6.3±3.3 (Lab-4) to 9.1±5.0 (Lab-1)
$\mu g/m^3$ in 10 labs, accounting for 20.6±4.8 % to 29.0±6.7 % of the $PM_{2.5}$ mass concentrations.
However, their contributions to total ions measured by each lab were not significantly different,
and ranged between 83.6±2.7% and 86.3±2.3%. The total ions summed to 24.3±4.9% (Lab-4)
to 33.8±7.1% (Lab-1) of $PM_{2.5}$. These results are comparable with those in another study in
Beijing which found that SNA accounted for 88% of total ions and 9-70% of $PM_{2.5}$
concentrations (Xu et al., 2019b). As shown in Table 2, the recoveries of most ions measured
by Lab-4 were < 100%, while those of Lab-1 were much higher, especially for major ions
(>100%). For Lab-6 which was also observed to have lower recoveries of ions such as the
lowest recoveries of $SO_4^{2-}$ (89.2%) and $NH_4^+$ (88.4%) in 10 labs; its SNA concentrations and
total ions accounted for 24.5±5.6 % and 28.7±6.0% of $PM_{2.5}$, respectively, the second lowest
among all labs. Hence, it is very important to run certified reference materials before any
sample analysis to ensure accuracy and good quality of data.

$K^+$ concentrations analysed by 10 labs followed a similar trend to $PM_{2.5}$ mass, except the
sample measured on a moderately polluted day (19[th]) by Lab-6, which is 2-3 times higher than





that measured by other labs. $F^-$ concentrations varied across 10 labs, but most of them shared
a similar trend. Some labs like Lab-8 did not follow the same trend due to reporting
undetectable $F^-$ concentrations. The $Na^+$ concentration on the least polluted day (20th) was
abnormally high in Lab-9, while its concentrations measured by other labs were generally low.
This could be due to $Na^+$ contamination during preparation or measurement of this sample, as
$Na^+$ concentrations in the rest of the samples measured by Lab-9 followed a similar trend as
that of other labs. The alkaline ions $Mg^{2+}$ and $Ca^{2+}$ are mostly originated from crustal dust and
mainly exist in coarse particles (Zou et al., 2018). Their mass concentrations varied
considerably due to their relatively low concentrations in aerosol samples and being sometimes
below the detection limits in some labs, such as Lab-6. Nevertheless, some labs like Lab-2, 3,
and 10 still followed a similar trend.

**3.2.2**    **Comparison with ToF-ACSM data**
As shown in Fig. 1, $Cl^-$, $NO_3^-$, $SO_4^{2-}$, $NH_4^+$ generally exhibited similar patterns, but due to
some outliers, such as $NO_3^-$ concentration measured by Lab-8 on the 16th , the median values
were selected to better represent the general levels and theoretical actual concentrations of ions
measured by different labs. The scatter plots of the median mass concentrations of $Cl^-$, $NO_3^-$,
$SO_4^{2-}$ and $NH_4^+$ in 10 labs (IC- $Cl^-$, $NO_3^-$, $SO_4^{2-}$ and $NH_4^+$) *versus* the non-refractory (NR)
species measured by the ToF-ACSM (ACSM- $Cl^-$, $NO_3^-$, $SO_4^{2-}$ and $NH_4^+$) are shown in Fig. 2.
The time series of IC and ACSM data at IAP and BUCT are plotted in Fig. S4.
Chloride is reported to arise mainly from biomass burning and coal combustion in China
(Zhang et al., 2016). Its average concentration in 10 labs correlated very well with ACSM-$Cl^-$
($R^2$=0.82 for IAP). However, IC-$Cl^-$ in IAP is 2-3 times higher than ACSM-$Cl^-$; this may be
due to the small contribution of $Cl^-$ to the overall mass spectrum which made it difficult to
quantify by ToF-ACSM (Allan et al., 2004). Additionally, the ACSM is incapable of measuring





$Cl^-$ in the form of KCl, as the ACSM only measures non-refractory $Cl^-$. Poor correlation of
chloride ($R^2$=0.21) was also discovered between two collocated ACSMs with a much larger set
of data points, while other NR species were observed with strong correlation ($R^2$>0.8) in
another study (Budisulistiorini et al., 2014), suggesting the quantification of chloride by ACSM
has large uncertainties.
Very good correlation between measured IC and ACSM data was found for $NO_3^-$ and $NH_4^+$
with $R^2$>0.9. The lab median value of $NO_3^-$ was very close to the ACSM-$NO_3^-$ from the same
sampling site- IAP, with a slope of 0.88 for IC-$NO_3^-$/ ACSM-$NO_3^-$, while that of BUCT was
only 0.57.    The slopes of IC-$NH_4^+$/ ACSM-$NH_4^+$ were 0.58 and 0.60 for IAP and BUCT,
respectively. Comparing IC-$NH_4^+$ to ACSM-$NH_4^+$, the absolute concentration of IC-$NH_4^+$
differed the most among all ions (42%), except $Cl^-$. Generally, ACSM-$NO_3^-$ and ACSM-$NH_4^+$
were higher than the median values of measured $NO_3^-$ and $NH_4^+$ concentrations in the 10 labs.
A potential reason is the high volatility of these species which leads to higher concentrations
in the online ACSM observations compared to the daily filter sample measurements due to
negative filter artefacts. It is also possible that the representative ions of ACSM-$NO_3^-$ and -
$NH_4^+$ could have significant interferences from other species in the mass spectrum, causing
large uncertainties even after correction for those interferences.
Sulfate, as another important component of atmospheric secondary inorganic aerosols, plays
an important role in the formation of haze (Wang et al., 2014; Yue et al., 2019). The correlation
coefficient ($R^2$) between the measured IC-$SO_4^{2-}$ and ACSM-$SO_4^{2-}$ was only 0.26 for IAP with
a slope of 0.54, but $R^2$ increased to 0.82 when excluding an outlier of the data on 23[rd]. The
correlation of IC-$SO_4^{2-}$ and ACSM-$SO_4^{2-}$ from BUCT was 0.84 ($R^2$) with a slope of 0.56.
Judging from the slopes, ACSM-$SO_4^{2-}$ and ACSM- $NH_4^+$ were similarly higher than the median
values of measured $SO_4^{2-}$ and $NH_4^+$ concentrations in this study. The NR species followed the



same trend as NR-PM$_{2.5}$, and chemical species measured through filter analysis also shared the
same trend as PM$_{2.5}$ measured in our study.
To summarize, SO$_4^{2-}$, NO$_3^-$, NH$_4^+$ from lab analysis generally correlated very well with the
ACSM data, but the absolute concentrations differ by up to 42%. Cl$^-$ from the two methods is
correlated but the concentration differ by more than 3 times. Crenn et al. (2015) reported the
uncertainties of NO$_3^-$, SO$_4^{2-}$, and NH$_4^+$ in ACSM analysis were 15%, 28%, and 36%. It appears
that Cl$^-$ is less accurate in online ACSM observations. NO$_3^-$ was comparable for the online data
and filter-based data, while SO$_4^{2-}$ and NH$_4^+$ in online data may be generally overestimated by
a similar factor. It should be noted that higher SO$_4^{2-}$ concentrations in online ACSM data could
potentially be due to ACSM not being able to separate organosulfate from sulfate. ACSM-NO$_3^-$,
-SO$_4^{2-}$ and -NH$_4^+$ were also reported to be higher (approximately 10-20%) than filter analysis
based NO$_3^-$, SO$_4^{2-}$ and NH$_4^+$ in another study (Sun et al., 2020).   Although the comparison
between IC and ACSM provided important information about the data from the two methods,
we recognize that we only have 8 data points here. Future studies should be carried out and
include more data points in order to comprehensively study the relationship between the online
ACSM data and filter-based data. We emphasize that it is essential that the filter-based
observations are robustly quality controlled before any ACSM and IC intercomparison.

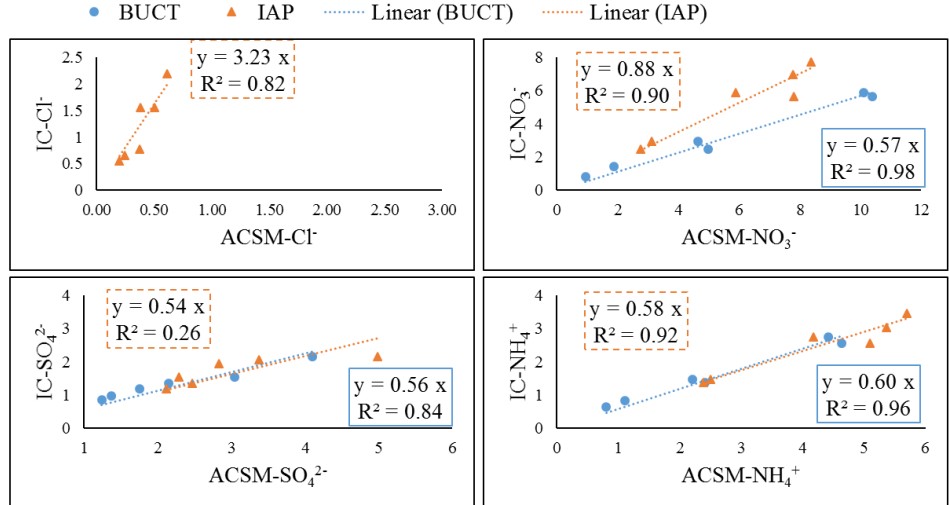

**Fig. 2.** Scatter plots of the median mass concentrations of $Cl^-$, $NO_3^-$, $SO_4^{2-}$ and $NH_4^+$ measured by 10 labs (IC- $Cl^-$, $NO_3^-$, $SO_4^{2-}$ and $NH_4^+$) *versus* the non-refractory (NR) chemical species from ACSM (ACSM- $Cl^-$, $NO_3^-$, $SO_4^{2-}$ and $NH_4^+$) from BUCT and IAP.

### 3.2.3 $NO_3^-/SO_4^{2-}$ ratios

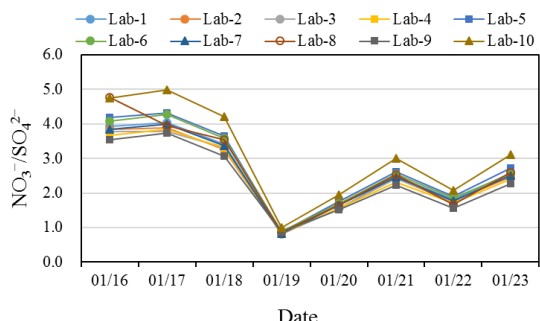

**Fig. 3.** Mass ratio of $NO_3^-/SO_4^{2-}$ during the study period in 10 labs.

Good agreement was observed for the mass ratios of $NO_3^-/SO_4^{2-}$ in most of the labs during the study period, which basically followed a similar trend as $PM_{2.5}$. On more polluted days, $NO_3^-/SO_4^{2-}$ ratios were obviously higher than less polluted days, suggesting the dominance of mobile source contributions over stationary sources during heavily polluted days.





### 3.3    Divergence and Correlation Analysis

As shown above, some ions like $Cl^-$, $NO_3^-$, $SO_4^{2-}$, $NH_4^+$ generally exhibited similar patterns,

but some of the ions varied significantly in different laboratories. Therefore, the Pearson's

correlation coefficient (R) and the coefficient of divergence (COD) were both calculated to

identify the uniformity and divergence of ionic concentrations measured by different labs. The

COD and R values of all ions for $Lab_j$/Lab-Median pairs are presented in Fig. 4. $Cl^-$, $NO_3^-$,

$SO_4^{2-}$, $NH_4^+$ and $K^+$ clearly showed high R values (>0.8) and low COD values (<0.2) in all

labs, suggesting the reliability of the measurement of these ions in different labs. However, $F^-$

and $Ca^{2+}$ in most labs was observed with higher COD values, and $Ca^{2+}$ was also found with

lower R, suggesting heterogeneity of $Ca^{2+}$ detection in different labs, which made this ion less

reliable. $Mg^{2+}$ was observed with good correlation (>0.7) between each lab and the Lab-Median,

but a higher COD was found between Lab-3, 5, 6 with the Lab-Median. Similarly, $Na^+$ was

also observed with good correlation (>0.7) between each lab and the Lab-Median, except Lab-

9, and a higher COD was found between Lab-5, 8 with the Lab-Median.

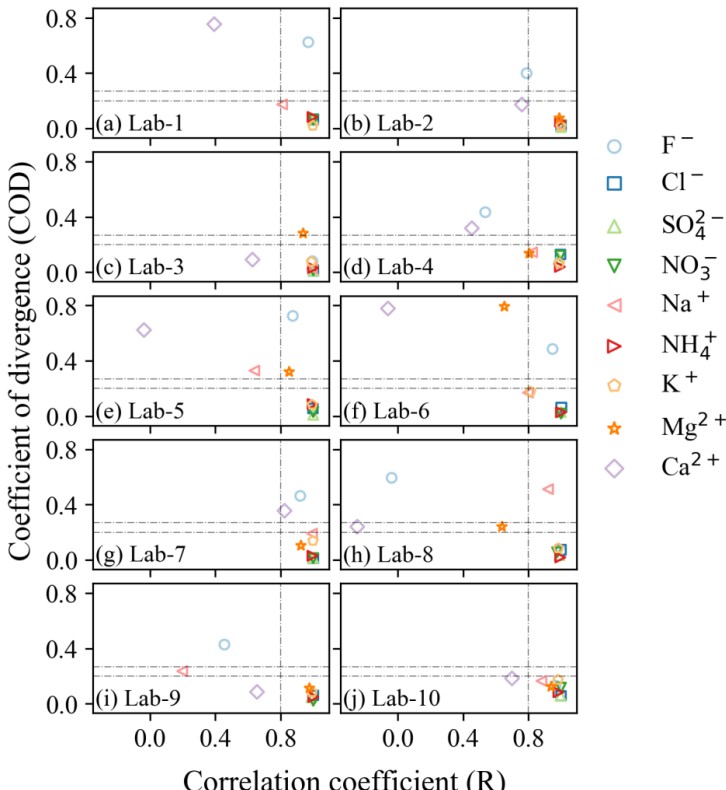

**Fig. 4.** Coefficient of divergence (COD) plotted against correlation coefficient (R) for all ions in each lab with the median ionic concentrations of 10 labs. (Note: vertical line indicates an R value of 0.8, and horizontal lines indicate COD values of 0.2 and 0.269, respectively).

### 3.4 Correction of Ion Concentrations by Recovery of CRM

The recovery of the certified reference materials was used to correct the ion concentrations in this study. The correction was conducted by dividing the measured ion concentrations by their corresponding recovery value. The coefficient of variation (CV) which can indicate the variance of data, was applied here to compare the variation of uncorrected/corrected ion concentrations among 10 labs. It was calculated as the standard deviation of ion concentrations measured by 10 labs divided by the mean and expressed in a percentage. A lower CV value indicates the closeness of data measured by 10 labs and reflects more precise results, while higher CV value reflects the opposite. As $F^-$, $Na^+$, $Mg^{2+}$ and $Ca^{2+}$ were undetectable in some





labs, only $Cl^-$, $SO_4^{2-}$, $NO_3^-$, $NH_4^+$ and $K^+$ were investigated and the results are shown in Table

384    4.


In Table 4, Lab-7 was excluded from the calculation of CV of both uncorrected and corrected
chloride, due to its poor repeatability. The CV of uncorrected chloride concentration in 8
samples varied between 11.7-19.3%, with an average of 14.3%. CV of corrected chloride
concentration in 8 samples varied between 10.4-17.0%, with an average of 12.6%. The
averaged CV decreased 1.7% for corrected chloride concentration. Small changes of CV were
observed during moderately polluted days (16[th], 17[th], 18[th]), but more obvious changes occurred
during non-haze days. These results suggested that certified reference materials can be used to
correct the $Cl^-$ concentrations for more accurate results, especially for less polluted samples.

The average CV of $SO_4^{2-}$ surprisingly increased from 9.8% for uncorrected to 10.9% for
corrected $SO_4^{2-}$ (Supplemental Table S1). However, when excluding Lab-3 from the
calculation, the averaged CV of uncorrected sulfate concentration was 10.3% and it
significantly decreased to 6.9% once corrected. Therefore, it is strongly recommended that
excessive recovery (>110%) with large variation should be avoided for the correction of $SO_4^{2-}$
concentrations. Better agreements of $NO_3^-$ and $K^+$ concentrations among 10 labs were also
observed after correction, as indicated by lower CV values for corrected samples. Similar to
other ions, the mean concentration of $NH_4^+$ of the 10 labs remained almost the same after
correction, but the CV of corrected samples increased from 12.5% to 13.2% after correction
(Supplemental Table S1). Nevertheless, it decreased 1.2% after correction when excluding
Lab-2 from the calculation, the $NH_4^+$ recovery of which was 135.0±6.0 %. The small change





of coefficient of variation here could be due to the high volatility of ammonia which leads to
differing results measured by different analytical procedures in labs.

To sum up, certified reference materials should be applied for the correction of the ion
concentrations. But the extreme recoveries with large inter-CRM variations should be avoided
from the corrections, as this may increase the uncertainty of measurements.

**Table 4.** Uncorrected and CRM-corrected ion concentrations (μg/m³) and their corresponding
coefficient of variations (CV/ %).

| | Uncorrected Mean (min-max) | CV/% | Corrected Mean (min-max) | CV/% | Uncorrected Mean (min-max) | CV/% | Corrected Mean (min-max) | CV/% |
|---|---|---|---|---|---|---|---|---|
| | Chloride | | | | Sulfate | | | |
| 2019/1/16 | 1.5 (1.2-1.8) | 11.7 | 1.5 (1.2-1.7) | 10.4 | 1.5 (1.1-1.7) | 11.3 | 1.6 (1.2-1.7) | 8.8 |
| 2019/1/17 | 2.2 (1.8-2.6) | 12.4 | 2.2 (1.7-2.6) | 11.3 | 2.0 (1.6-2.3) | 9.7 | 2.0 (1.7-2.2) | 6.0 |
| 2019/1/18 | 1.5 (1.2-1.8) | 11.9 | 1.5 (1.2-1.8) | 11.2 | 2.0 (1.6-2.4) | 10.2 | 2.1 (1.7-2.3) | 7.3 |
| 2019/1/19 | 0.2 (0.1-0.3) | 19.3 | 0.2 (0.2-0.2) | 16.8 | 1.0 (0.9-1.1) | 7.9 | 1.0 (1.0-1.1) | 4.5 |
| 2019/1/20 | 0.3 (0.2-0.4) | 19.0 | 0.3 (0.3-0.4) | 17.0 | 0.9 (0.8-1.1) | 10.7 | 0.9 (0.8-1.0) | 6.7 |
| 2019/1/21 | 0.6 (0.5-0.8) | 12.6 | 0.6 (0.5-0.7) | 11.0 | 1.2 (1.1-1.4) | 8.7 | 1.2 (1.1-1.3) | 4.7 |
| 2019/1/22 | 0.5 (0.4-0.7) | 13.4 | 0.5 (0.4-0.6) | 11.3 | 1.4 (1.0-1.6) | 12.5 | 1.4 (1.1-1.6) | 8.8 |
| 2019/1/23 | 0.8 (0.5-0.9) | 13.9 | 0.8 (0.6-0.8) | 12.0 | 2.2 (1.7-2.5) | 11.6 | 2.3 (1.8-2.4) | 8.5 |
| Average | | 14.3 | | 12.6 | | 10.3 | | 6.9 |
| | Nitrate | | | | Ammonium | | | |
| 2019/1/16 | 6.1 (4.1-8.0) | 16.5 | 6.1 (4.5-8.3) | 15.2 | 2.7 (2.1-3.2) | 12.7 | 2.7 (2.1-3.2) | 12.8 |
| 2019/1/17 | 8.0 (6.1-9.8) | 13.1 | 8.0 (6.7-8.9) | 7.8 | 3.6 (2.6-4.5) | 14.9 | 3.6 (2.9-4.2) | 12.1 |
| 2019/1/18 | 7.1 (5.3-8.3) | 12.1 | 7.1 (5.7-7.9) | 8.4 | 3.1 (2.7-3.8) | 10.8 | 3.2 (2.6-3.8) | 10.2 |
| 2019/1/19 | 0.9 (0.7-0.9) | 8.9 | 0.9 (0.8-1.0) | 7.3 | 0.6 (0.5-0.8) | 11.7 | 0.6 (0.6-0.7) | 9.4 |
| 2019/1/20 | 1.5 (1.2-1.7) | 9.8 | 1.5 (1.3-1.6) | 7.0 | 0.8 (0.6-1.0) | 13.1 | 0.8 (0.7-1.1) | 13.3 |
| 2019/1/21 | 3.0 (2.4-3.4) | 9.4 | 3.0 (2.7-3.3) | 5.9 | 1.5 (1.1-1.7) | 12.1 | 1.5 (1.3-1.7) | 9.7 |
| 2019/1/22 | 2.4 (1.8-2.9) | 12.3 | 2.5 (2.0-2.6) | 7.9 | 1.3 (1.0-1.5) | 12.3 | 1.3 (1.1-1.6) | 11.8 |
| 2019/1/23 | 5.7 (4.0-6.8) | 13.6 | 5.7 (4.4-6.4) | 9.6 | 2.5 (2.0-3.0) | 13.7 | 2.6 (2.1-3.0) | 12.6 |
| Average | | 12.0 | | 8.6 | | 12.7 | | 11.5 |
| | Potassium | | | | | | | |
| 2019/1/16 | 0.3 (0.2-0.5) | 19.8 | 0.4 (0.3-0.5) | 16.2 | | | | |
| 2019/1/17 | 0.5 (0.3-0.6) | 15.6 | 0.5 (0.4-0.7) | 14.9 | | | | |
| 2019/1/18 | 0.3 (0.3-0.4) | 14.1 | 0.4 (0.3-0.5) | 10.8 | | | | |
| 2019/1/19 | 0.1 (0.1-0.3) | 48.5 | 0.1 (0.1-0.3) | 47.7 | | | | |
| 2019/1/20 | 0.1 (0.1-0.2) | 31.4 | 0.2 (0.1-0.3) | 29.7 | | | | |
| 2019/1/21 | 0.2 (0.1-0.3) | 20.9 | 0.2 (0.2-0.3) | 17.0 | | | | |
| 2019/1/22 | 0.2 (0.1-0.3) | 20.6 | 0.2 (0.1-0.3) | 17.8 | | | | |
| 2019/1/23 | 0.3 (0.2-0.3) | 25.3 | 0.3 (0.2-0.4) | 21.3 | | | | |





| | | |
|---|---|---|
| Average | 24.5 | 21.9 |

*Lab-2, 3 and 7 were excluded for calculating CV% of ammonium, sulfate and chloride, respectively.*

### 417  3.5    Aerosol Acidity

In this study, aerosol acidity was evaluated applying three different parameters: Anion and
Cation Equivalence Ratio, ion-balance and in situ acidity. Ion-balance was calculated by
subtracting equivalent cations from anions (Zhang et al., 2007), while in-situ aerosol acidity
was represented by pH or the concentration of free $H^+$ in the deliquesced particles under
ambient conditions. In situ aerosol pH can be estimated from various thermodynamic models,
for example, SCAPE, GFEMN, E-AIM and ISORROPIA (He et al., 2012; Pathak et al., 2009;
Yao et al., 2006). In situ aerosol acidity is most likely to influence the chemical behavior of
aerosols (He et al., 2012). Ion-balance is widely used to indicate the neutralization status of
aerosols with the equivalent ratios of anions/cations in a relative way (Sun et al., 2010; Takami
et al., 2007; Chou et al., 2008). It is noteworthy that ion-balance and in-situ aerosol acidity
estimations are empirical approaches which are strongly dependent on the selection of ion
species.

### 431  3.5.1    Anion and Cation Equivalence Ratio

The ratio of the anion molar equivalent concentrations to the cation molar equivalent
concentrations (AE/CE) can be applied to reflect the potential aerosol acidity (Meng et al.,
2016; Zou et al., 2018). In this study, AE and CE were calculated as:
$$AE = [SO_4^{2-}/96] \times 2 + [NO_3^-/62] + [Cl^-/35.5] + [F^-/19] \tag{4}$$
$$CE = [NH_4^+/18] + [Na^+/23] + [K^+/39] + [Mg^{2+}/24] \times 2 + [Ca^{2+}/40] \times 2 \tag{5}$$



AE represents the equivalent concentrations of all anions; and CE denotes all cations equivalent
concentrations.
**Table 5.** Anion and cation equivalent ratios (AE/CE) among 10 laboratories.

|  | Lab-1 | Lab-2 | Lab-3 | Lab-4 | Lab-5 | Lab-6 | Lab-7 | Lab-8 | Lab-9 | Lab-10 |
|---|---|---|---|---|---|---|---|---|---|---|
| 2019/1/16 | 1.02 | 1.01 | 1.02 | 0.81 | 1.26 | 0.93 | 1.03 | 1.18 | 0.93 | 1.43 |
| 2019/1/17 | 1.00 | 1.02 | 1.01 | 0.85 | 1.25 | 0.93 | 0.87 | 1.07 | 0.96 | 1.59 |
| 2019/1/18 | 1.03 | 1.03 | 1.04 | 0.84 | 1.26 | 0.96 | 1.03 | 1.14 | 0.95 | 1.28 |
| 2019/1/19 | 0.99 | 0.79 | 0.97 | 0.85 | 1.11 | 0.65 | 0.99 | 0.90 | 0.98 | 1.15 |
| 2019/1/20 | 1.00 | 0.80 | 0.96 | 0.85 | 1.14 | 0.82 | 1.00 | 0.98 | 0.83 | 1.08 |
| 2019/1/21 | 1.03 | 0.78 | 1.03 | 0.80 | 1.14 | 0.85 | 1.04 | 1.02 | 0.90 | 1.12 |
| 2019/1/22 | 1.04 | 0.79 | 1.04 | 0.80 | 1.16 | 0.90 | 0.97 | 0.91 | 0.93 | 1.09 |
| 2019/1/23 | 1.02 | 0.98 | 1.05 | 0.80 | 1.15 | 0.95 | 0.84 | 1.00 | 0.94 | 1.48 |


As presented in Table 5, the AE/CE ratio of all samples were compared among 10 labs. The
ratios in Lab-1 and Lab-3 were close to unity. The ratios in Lab-5 and Lab-10 were above 1,
indicating the deficiency of cations to neutralize all anions, while that was the contrary of Lab-
4, 6 and 9. In Table 2, the recoveries of major cations ($Na^+$, $NH_4^+$, $K^+$) were <100% and much
lower than those of the major anions ($Cl^-$, $NO_3^-$, $SO_4^{2-}$) in Lab-5 and 10, which may have caused
lower cation concentrations than their real concentrations and a constant higher ratio of AE/CE.
For Lab-9, the recoveries of all ions were very close to 100%, except $NH_4^+$ which was found
with a recovery of >110%. Therefore, AE/CE < 1 of all samples measured by Lab-9 could be
the result of overestimation of ammonium. Similarly, in addition to an ammonium recovery
of >110%, generally lower anion recoveries than cations were reported by Lab-4, which may
explain AE/CE < 1 in all samples measured by this lab as well. The other three labs (Lab-2, 7
and 8) were found with various AE/CE ratios with both >1 and <1 values; moderately polluted
days were generally observed with a higher ratio of AE/CE. These results indicate that AE/CE
ratios bear large uncertainties from different labs. Stricter quality control measures should be
adopted if applying AE/CE ratios to evaluate aerosol acidity.






### 3.5.2 Ion Balance

The calculation of ion balance is an alternative way to evaluate the aerosol acidity (Han et al.,
2016; He et al., 2012). Three methods were listed below for the calculation of ion balance in
this study:
Method 1: $IB = 2[SO_4^{2-}] + [NO_3^-] - [NH_4^+]$ (6)
Method 2: $IB = 2[SO_4^{2-}] + [NO_3^-] + [Cl^-] - [NH_4^+] - [Na^+] - [K^+]$ (7)
Method3: $IB = 2[SO_4^{2-}] + [NO_3^-] + [Cl^-] - [NH_4^+] - [Na^+] - [K^+] - 2[Mg^{2+}] - 2[Ca^{2+}]$

466 (8)

In Method 1, only $SO_4^{2-}$, $NO_3^-$ and $NH_4^+$ were applied for the calculation (Tian et al., 2017),
assuming that these three ions and $H^+$ alone control $PM_{2.5}$ acidity (Ziemba et al., 2007). $SO_4^{2-}$,
$NO_3^-$ and $NH_4^+$ were also used in other studies to assess aerosol acidity. For example, the mole
charge ratio of $NH_4^+$ to the sum of $SO_4^{2-}$ and $NO_3^-$ was applied to represent aerosol acidity
(Chandra Mouli et al., 2003; Wang et al., 2019). $SO_4^{2-}$, $NO_3^-$ and $NH_4^+$ were selected because
they contributed approximately 90% of the total ionic species in fine aerosols and play
predominant roles in controlling aerosol acidity (Zhou et al., 2012). Salt ions $Na^+$, $K^+$ and $Cl^-$
were added for the calculation in Method 2. Based on this calculation, $Mg^{2+}$ and $Ca^{2+}$ were
added in Method 3 to include the effects of crustal dust on aerosol acidity (Huang et al., 2014).
The ion balance of all labs varied applying different methods, especially for the first three
heavily polluted days, as shown in Fig. 5. Positive ion balance values indicated a deficiency of
cations to neutralize anions, while negative values implied an excess of cations to neutralize
anions. Lab-10 showed the highest variation among all labs; when excluding Lab-10, the results
of the other 9 labs agreed very well, with most of the values below 0, suggesting sufficient
ammonium to neutralize sulfate and nitrate. By applying Method 1, comparable results were



found. The average ion balance values in all samples were consistent in Lab-1, 2, 6, 7, 9 (0.02
μmol/m³). When adding more ions in the calculation by adopting Methods 2 and 3, poorer
agreement among all labs was exhibited. Therefore, it seems more consistent to indicate the
relative ion-balanced aerosol acidity among different samples by Method 1, as SNA were the
most abundant ions in atmospheric aerosols and their concentrations measured by different labs
showed good agreement (Fig. 1). This method could reduce the large discrepancy of ion
balance results calculated by adding other ions from the different labs, as their concentrations
varied largely in different labs due to varying detection limits.

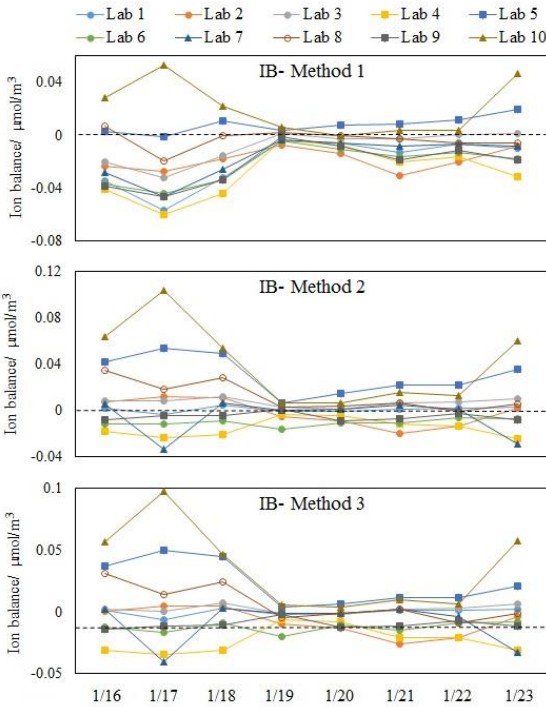


**Fig. 5** Ion balance in all labs applying different methods (negative values reflect the excessive cations
to neutralize anions)

**3.5.3    Aerosol pH using ISORROPIA-II**
A thermodynamic equilibrium model- ISORROPIA-II was applied to estimate the in-situ
aerosol acidity. This was run only in forward mode, as the results from the use of reverse mode
(using only particle phase composition) are reported to be unreliable (Song et al., 2018).   The
only gas phase data were for ammonia, but this introduces little error as concentrations of $HNO_3$
and HCl are likely to be very low in this high ammonia environment (Song et al., 2018).
The inputs include aerosol-phase $Cl^-$, $SO_4^{2-}$, $NO_3^-$, $Na^+$, $NH_4^+$, $K^+$, $Mg^{2+}$, $Ca^{2+}$ and gas-phase
$NH_3$ concentrations. The daily ammonia concentrations during the study period ranged from
$13.9\pm0.6$ to $20.1\pm0.7$ ppb with an average of $17.2\pm2.2$ ppb. Mean $NH_3$ concentrations during
moderately polluted and non-haze days were $19.6\pm0.6$ and $15.9\pm1.5$ ppb, respectively. Daily
temperature ranged between -4.4°C to 4.3°C with an average of 1.0°C and RH ranged from
13.8% to 40.1% with a mean value of 22.4%. The aerosol pH was calculated for all samples
by the model, as well as aerosol water content (AWC. Table S3), details of the calculation of
pH and AWC can be found elsewhere (Liu et al., 2017b; Masiol et al., 2020). The calculated
aerosol pH results of 10 labs are presented in Fig. 6.

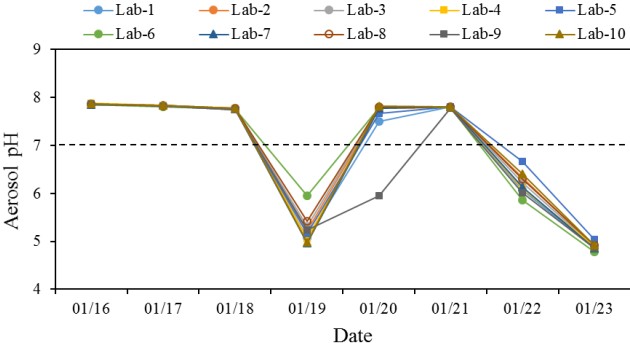


**Fig. 6.** Aerosol pH estimated by ISORROPIA-II using ions and ammonia in 10 labs from 16th to 23rd
January 2019.

The computed aerosol pH during the study period generally exhibited good agreement among
10 labs. Lab-6 was observed with higher pH and lower ion balance than other labs on the 19th,
which could be mainly due to the 2-3 times higher $K^+$ concentration measured by Lab-6 on that
day (Fig. 1), while other ions measured by this lab were more comparable with other labs. The
aerosol pH on 3 moderately polluted days was above 7, indicating an alkaline nature of aerosols





during these days. This result is consistent with the discussion mentioned above that ion
balance estimated by Method 1 was below 0 as more $NH_4^+$ neutralizes $NO_3^-$ and $SO_4^{2-}$.
Excellent agreement among the 10 labs for the aerosol pH during these moderately polluted
days was also found. Non-haze days, especially the least polluted day on 20[th], showed higher
variation among the different labs. The calculated pH of 9 labs mostly fall on the same side of
the neutralization line (pH=7), and only lab-9 on 20[th] falls onto a different side of the pH=7
line from the other labs.
$NH_3$ is the main driving factor affecting aerosol pH and leads to the more alkaline nature of
aerosols. Wang et. al (2020) also reported that the high concentration of total ammonium
(gas+aerosol) was likely an important factor causing lower aerosol acidity of fine particles
during a severe haze period in Henan province, China. It is also confirmed in another study that
ammonia played an important role in influencing aerosol pH during winter haze period in
northern China (Song et al., 2018).

**4.   SUMMARY AND RECOMMENDATIONS**
Despite use of variable methods and instruments for measuring ion concentrations, data from
all the participating labs show a reasonably good agreement in the overall trend for major ions
like chloride, sulfate, nitrate, and ammonium. The coefficients of divergence of these ions
across 10 labs were lower than 0.2 and the correlation coefficients were higher than 0.8,
suggesting a reasonably high reliability of measuring major ions by IC in different labs.
However, the inter-lab difference can be as high as 30% if excluding the two extreme values
for each day, and reached up to 100% in extreme cases if including all data. Furthermore, ions
like $F^-$, $Mg^{2+}$, $K^+$ and $Ca^{2+}$ were observed with large variations in different labs, which may be
due to their relatively low concentrations in the atmosphere. Good correlations were found for





non-refractory ion species measured by ACSM with those in our study. However, the absolute
mass levels were quite different, which may be due to the interference of other ions in mass
spectra and in some cases, the capture of semi-volatile species by the ACSM. Due to the limited
number of datapoints in the present study, further investigations on larger datasets are
necessary to confirm these findings. Certified reference materials were applied to show the
detection accuracy of IC measurement in the 10 laboratories. By comparing the coefficient of
variation of samples among 10 labs before and after correction by the recovery of CRM, we
emphasize the importance of using certified reference materials for quality control for future
ionic species analysis.

Aerosol acidity was studied through the investigation of ion-balance based acidity and in-situ
acidity. Firstly, the ratios of anion equivalent concentrations to cation equivalent concentrations
(AE/CE) varied significantly in different labs, which could be attributed to measurement errors,
as supported by the different recoveries of ions in CRM. Secondly, by calculating the ion
balance, Method 1 which only applied SNA for the calculation, was more consistent in most
labs. Poor agreement of acidity estimation was observed in all labs when adding other ions like
$Ca^{2+}$ and $Mg^{2+}$. Finally, ISORROPIA-II was applied for estimating in-situ aerosol acidity by
calculating aerosol pH in forward (gas+aerosol phases as input) mode. The results showed a
similar trend between labs and exhibited a good agreement. This indicates that, if including
gaseous pollutant equilibrium in the ISORPIA II model, the estimated aerosol pH is more
consistent even if there are relatively large differences in the measured concentrations of ions.

Based on this analysis and our experience, we recommend that:





1. There are substantial inter-lab uncertainties in both the aerosol major and minor ions measured by ion chromatography from the filters. Literature data should be treated with this uncertainty in mind.

2. The ion-balance approach bears large uncertainty and thus should be used with caution for estimating aerosol acidity. Instead, in situ aerosol pH may be used to represent acidity, and can be calculated from thermodynamic model considering gas-aerosol equilibrium (e.g., $NH_4^+$ and $NH_3$)

3. Poor consistency of ion-balanced acidity in 10 labs, but good agreement of aerosol pH when applying the ISORROPIA-II model, suggests that measurement errors were not that important when including $NH_3$ for the determination of aerosol pH. Hence, we recommend $NH_3$ should be used in future aerosol pH investigations.

4. Certified reference materials should be used on a regular basis to assess the accuracy of the measurement method and ensure reliable measurements.

5. Actual aerosol observations should be corrected for CRM recoveries. But the recovery of ions with poor repeatability should be not be used for correction, as it will cause a larger discrepancy. IC performance should be checked and improved when ions exhibit poor repeatability.

6. The recovery of ammonium varied significantly among 10 labs (88.4-135.0%). As ammonium is highly volatile, it is recommended that stock solutions which are used for the preparation of calibration standards should be freshly prepared to ensure the detection accuracy.

7. As the ions like $Mg^{2+}$ and $Ca^{2+}$ were not detectable in some labs due to low concentrations, it is recommended that labs should consider improving their detection limits and /or correspondingly increase the size of filter cuts for future analysis.



8.  Robust quality control processes should be put in place to avoid contamination,

590       particularly for those ions with low concentrations, such as $K^+$ and $Na^+$.

9.  Some batches of commercial quartz filters may be contaminated with $Na^+$ and $PO_4^{3-}$, and

592       thus testing each batch of blank filters is necessary before any field sampling (data not

593       shown here).

10.  Ionic concentration from ACSM observations should be calibrated although the observed

595       trend is robust.


**ACKNOWLEDGEMENTS**
This research was funded by the Natural Environment Research Council (Grant Nos.
NE/S00579X/1, NE/N007190/1, NE/R005281/1). We would like to thank all researchers for
carrying out the technical work and providing the relevant data. We appreciate the support from
all participating laboratories.

*Data availability.* The data in this article are available from the corresponding author upon
request.

*Author contributions.* ZS conceived the study after discovering large inter-lab variability in
water-soluble inorganic ions from offline and online methods. JX prepared the paper with the
help of ZS, RMH and all co-authors. JX, LW, QZ, CZ, XY, DC, WJL, MW, HT, LiL, ST,
WRL, JW, GS, YH, SS, CP, YC, FY, AM, DD, SJS, IA, and JFH conducted the laboratory
analysis. SS supported the aerosol pH calculation. CS supported the calculation of coefficient
of divergence. YLS, LuL, FZ, KRD, CY, YL, MK provided the ACSM data and YLS supported
the interpretation of the ACSM data. BG provided the $NH_3$ data.

*Competing interests.* The authors declare that they have no conflict of interest.



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
