# Peer review of "An inter-laboratory comparison of aerosol inorganic ion measurements by Ion"

_Atmospheric Measurement Techniques, 2020_

## Referee Comment (RC1) · Anonymous Referee #1 · 19 May 2020

Review of "An inter-laboratory comparison of aerosol in organic ion measurements by Ion Chromatography: implications for aerosol pH estimate" by J. Xu et al.

**General Comments:**
This manuscript presents an intercomparison of IC measurements made by 10 independent research laboratories spanning three countries. The analysis includes ambient PM filter extracts and certified reference material (CRM). The ambient filter-IC results are also compared to two different ACSMs, revealing some interesting trends. Ultimately, the results are analyzed to understand how differences in filter-IC measurements produce different aerosol pH values modeled with ISORROPIA. In my opinion, this is a novel and quite significant study. The manuscript is well-organized and the writing is clear. There are a few issues that need to be addressed before publication, these are detailed below.

**Specific Comments:**
The first major issue relates to Section 3.5.1 ("Anion and Cation Equivalence Ratio") and Section 3.5.2 ("Ion Balance"). The authors are referred to several recent papers that have analyzed the use of these proxies for aerosol acidity (e.g., Guo et al., 2015; Hennigan et al., 2015). The findings are summarized in a recent review on aerosol acidity (Pye et al., 2020). In short, these methods are flawed representations of particle acidity in most environments, and it is advised that they not be used. I suggest removing these two sections, or at the very least, a major revision of these sections is required to accurately reflect the updated assessment of their inability to represent particle acidity.

Another major issue is with Section 3.4 and the method of using the CRM "recoveries". From what I understand, these are not recoveries in the traditional sense that the word is used with filter measurements (e.g., material spiked onto the filter, then measured in the filter extract), but it is instead more like a QA standard that one would run with a batch of samples. It seems that the "correction" applied in Section 3.4 is actually more like a single-point calibration. The justification seems to be that the CRM solutions may have been prepared more recently than a lab's calibration standards, allowing less time for artifacts to develop from contamination or volatilization (of water or analytes). However, it is presumed that each of the labs, themselves, used a CRM for their calibrations, though this is never stated. If true, then perhaps the result here speaks more to guidance on the frequency of calibrations. Also, it is good analytical chemistry protocol to run blanks and QA standards regularly interspersed with sample analyses. The QA standards should be made from CRM, as well. There are typically pre-defined limits of acceptable performance, if the QA standards fall within this predefined accuracy, there is no adjustment to the sample results. To summarize: each lab's calibration procedures should be detailed (likely in the SI); as should their QA/QC procedures for running a batch of filter samples. If it is found that a lab's calibration is off, then adjustments are warranted (again the QA standards should signal this), but if QA standards fall within a predetermined accuracy, then a "recovery" correction is questionable.

While not the central focus of the manuscript, the differences between the ICs and the ACSMs are significant and deserve more attention. The explanation in lines 312-314 seems highly

unlikely (volatilization from filters) since sulfate had very similar behavior.  Two ideas that were not discussed but which could have contributed to the discrepancies are: 1) differences in the performance of the two $PM_{2.5}$ cut-point selectors, which could lead to different transmission of particles in the 2-3 µm range, and 2) the collection efficiency applied for the ACSMs.  On the second point, the collection efficiency is often assumed to be 0.5, and is applied to an entire data set even though this changes with aerosol composition and meteorological conditions.  Therefore, an applied CE of 0.5 when the actual CE was closer to 1 would also produce the observed results.  Further, line 339-340 states that "it is essential that the filter-based observations are robustly quality controlled before any ACSM and IC intercomparison," but what about robust quality control of the ACSM measurements?  What was done for this study, and what improvements should be made?

**Technical Corrections:**

- Line 48: suggest "factor of three" instead of "3 times"
- Line 69: suggest deleting the second 'to' in this sentence
- Line 77-78: sentence awkward as written, suggest rephrasing
- Line 84: suggest replacing "nowadays" with "at present" (or similar)
- Line 183: change to "COD value equals to…"
- The COD value of 0.269 (line 187-189, Figure 4) seems highly arbitrary. There really is no discussion about the 0.20 vs. 0.269 – I suggest picking only one.
- Line 320: was an statistical outlier test actually performed? Be careful excluding data without reason, especially for a smaller *n* like this study.
- Line 328: suggest "factor of three" instead of "3 times"
- Suggest adding 1:1 lines to Figure 2
- Section 3.2.3 is quite short, and could be omitted, or incorporated into another section. Fig. 3 does not add much of substance to the overall discussion.
- Line 523-524: what is the hypothesized reason for the highly different pH value from Lab-9's data on this date?
- Line 565: I disagree that there are "substantial" uncertainties, especially in the major ions measured by IC.  The uncertainties are quantifiable, and they seem to be in line with previously published values.  For the "minor" ions, the threshold for major/minor is never stated, but clearly this is true for $Ca^{2+}$, while most of the others seemed to perform quite well.
- Line 582: yes, but I would note here that the median ammonium recovery was close to 100%.

**References:**
Guo, H., Xu, L., Bougiatioti, A., Cerully, K. M., Capps, S. L., Hite Jr., J. R., Carlton, A. G., Lee, S.-H., Bergin, M. H., Ng, N. L., Nenes, A., and Weber, R. J.: Fine-particle water and pH in the southeastern United States, Atmos. Chem. Phys., 15, 5211–5228, https://doi.org/10.5194/acp-15-5211-2015, 2015.

Hennigan, C. J., Izumi, J., Sullivan, A. P., Weber, R. J., and Nenes, A.: A critical evaluation of proxy methods used to estimate the acidity of atmospheric particles, Atmos. Chem. Phys., 15, 2775– 2790, https://doi.org/10.5194/acp-15-2775-2015, 2015.

Pye, H. O. T., Nenes, A., Alexander, B., Ault, A. P., Barth, M. C., Clegg, S. L., Collett Jr., J. L., Fahey, K. M., Hennigan, C. J., Herrmann, H., Kanakidou, M., Kelly, J. T., Ku, I.-T., McNeill, V. F., Riemer, N., Schaefer, T., Shi, G., Tilgner, A., Walker, J. T., Wang, T., Weber, R., Xing, J., Zaveri, R. A., and Zuend, A.: The acidity of atmospheric particles and clouds, Atmos. Chem. Phys., 20, 4809–4888, https://doi.org/10.5194/acp-20-4809-2020, 2020.

---

## Author Comment (AC1) · 20 May 2020

The comment was uploaded in the form of a supplement:
https://www.atmos-meas-tech-discuss.net/amt-2020-156/amt-2020-156-AC1-supplement.pdf

---

## Referee Comment (RC2) · Anonymous Referee #2 · 7 Jun 2020

The authors conduct an inter-laboratory comparison of off-line aerosol inorganic ion measurement by IC and show the measurement uncertainty of inorganic ions from different labs, as well as the influence of uncertainty in measuring WSII on the pH calculation. Generally, it is an interesting work to show us how important the QA/QC is. But more deep analysis is needed before it can be published in AMT.

1. My 1st concern is that the authors used large space to show the differences in the WSIs measurement, but did not discuss on the possible reasons.

2. The calculation of aerosol pH using ISORROPIA-II is dependent on gas-phase NH3 too much, which is not an easy species to measure. For most cases, especially measurement based on filter sampling, NH3 would not be measured simultaneously.

3. Fig. 6: There were 5 of the 8 samples showed a pH higher than 7. It is, to me, a bit too high. According to my best knowledge, except for dust samples, the pH of most aerosols should be lower than 7. Did the authors measure the pH of sample solutions before IC measurement? Were the 5 samples alkaline?

4. Since the same concentration of NH3 was used in ISORROPIA-II to estimate the pH of aerosol, I somehow doubt the similarity of estimated pH among different labs were due to the same input of gas-phase NH3 (As mentioned in comment 2).

5. Section 3.2.3: I don't think this part is relevant to the topic of this work. Suggest to omit it. 6. Similar to the last comment, I don't understand why the authors make a comparison with ACSM. It is a bit out of the scope of this work.

7. The figure quality needs to be improved.

---

## Referee Comment (RC3) · Anonymous Referee #3 · 24 Jun 2020

Please see attached referee report.

Please also note the supplement to this comment:
https://www.atmos-meas-tech-discuss.net/amt-2020-156/amt-2020-156-RC3-supplement.pdf

---

## Author Comment (AC3) · 21 Aug 2020

The comment was uploaded in the form of a supplement:
https://amt.copernicus.org/preprints/amt-2020-156/amt-2020-156-AC3-supplement.zip
* * *

---

## Author Comment (AC4) · 21 Aug 2020

The comment was uploaded in the form of a supplement:
https://amt.copernicus.org/preprints/amt-2020-156/amt-2020-156-AC4-supplement.zip

---

## Author Comment (AC2)

**Point-by-Point Response to Reviewers' Comments**

**Manuscript Ref: amt-2020-156**

Title: An inter-laboratory comparison of aerosol in organic ion measurements by Ion Chromatography: implications for aerosol pH estimate

Journal: Atmospheric Measurement Techniques

**Comments from Reviewer #1**

**General Comments:**

There are a few issues that need to be addressed before publication, these are detailed below.

**General response:** Thanks for the valuable comments. Please see our point-to-point response below.

**Specific Comments:**

**Comments 1:**

The first major issue relates to Section 3.5.1 ("Anion and Cation Equivalence Ratio") and Section 3.5.2 ("Ion Balance"). The authors are referred to several recent papers that have analyzed the use of these proxies for aerosol acidity (e.g., Guo et al., 2015; Hennigan et al., 2015). The findings are summarized in a recent review on aerosol acidity (Pye et al., 2020). In short, these methods are flawed representations of particle acidity in most environments, and it is advised that they not be used. I suggest removing these two sections, or at the very least, a major revision of these sections is required to accurately reflect the updated assessment of their inability to represent particle acidity.

**Response:** We agree with the reviewer that "Anion and Cation Equivalence Ratio" and "Ion Balance" are flawed representations of the aerosol acidity. This is the conclusion that we can draw from our discussion in Section 3.5.1 and Section 3.5.2, which is also in good agreement with the papers that the reviewer mentioned. We consider that it is of value to include these sections to provide further support to the conclusions to these recent papers. As suggested, we have revised these sections and addressed the uncertainties of these estimations.

We added the discussion below:

"Our results suggest AE/CE and Ion Balance are flawed representations of particle acidity, which are not recommended for the evaluation of aerosol acidity. This is also consistent with the conclusions from previous studies (Hennigan et al., 2015; Guo et al., 2015; Pye et al., 2020). ISORROPIA-II gives more consistent aerosol pH values

among different laboratories. But there are uncertainties within this calculation: 1) RH during some periods in this study was relatively low (around 20%), and as a result, aerosol water content is very low. Under such conditions, ions are mostly existed in solid phase. Hence, pH of aerosols with very low RH may not be reliable; 2) the calculation of AWC only considered for inorganics in this study. Water associated with organics also contribute to AWC. For example, Guo et al. (2015) indicated that it accounts for 29-39% of total  $PM_{2.5}$  water in southeastern United States." Please see **line 555-564** in the revised manuscript.

"The ion-balance approach bears large uncertainty and thus should be used with caution for estimating aerosol acidity." in original manuscript line 568-569 has been revised as "The ion-balance approach is not recommended for estimating aerosol acidity due to its large uncertainty." in the revised manuscript line 608-609.

**Added references:**

Guo, H., Xu, L., Bougiatioti, A., Cerully, K., Capps, S., Hite, J., Carlton, A. M., Lee, S.-H., Bergin, M., Ng, N., Nenes, A., and Weber, R.: Fine-particle water and pH in the southeastern United States, ATMOSPHERIC CHEMISTRY AND PHYSICS, 15, 5211-5228, 10.5194/acp-15-5211-2015, 2015.

Hennigan, C. J., Izumi, J., Sullivan, A. P., Weber, R. J., and Nenes, A.: A critical evaluation of proxy methods used to estimate the acidity of atmospheric particles, Atmos. Chem. Phys., 15, 2775-2790, 10.5194/acp-15-2775-2015, 2015.

Pye, H. O. T., Nenes, A., Alexander, B., Ault, A. P., Barth, M. C., Clegg, S. L., Collett Jr, J. L., Fahey, K. M., Hennigan, C. J., Herrmann, H., Kanakidou, M., Kelly, J. T., Ku, I. T., McNeill, V. F., Riemer, N., Schaefer, T., Shi, G., Tilgner, A., Walker, J. T., Wang, T., Weber, R., Xing, J., Zaveri, R. A., and Zuend, A.: The acidity of atmospheric particles and clouds, Atmos. Chem. Phys., 20, 4809-4888, 10.5194/acp-20-4809-2020, 2020.

**Comments 2:**

Another major issue is with Section 3.4 and the method of using the CRM "recoveries". From what I understand, these are not recoveries in the traditional sense that the word is used with filter measurements (e.g., material spiked onto the filter, then measured in the filter extract), but it is instead more like a QA standard that one would run with a batch of samples. It seems that the "correction" applied in Section 3.4 is actually more like a single-point calibration. The justification seems to be that the CRM solutions may have been prepared more recently than a lab's calibration standards, allowing less time for artifacts to develop from contamination or volatilization (of water or analytes). However, it is presumed that each of the labs, themselves, used a CRM for their calibrations, though this is never stated. If true, then perhaps the result here speaks more to guidance on the frequency of calibrations. Also, it is good analytical chemistry protocol to run blanks and QA standards regularly interspersed with sample analyses. The QA standards should be made from CRM, as well. There are typically pre-defined limits of acceptable performance, if the QA standards fall within this predefined

accuracy, there is no adjustment to the sample results. To summarize: each lab's calibration procedures should be detailed (likely in the SI); as should their QA/QC procedures for running a batch of filter samples. If it is found that a lab's calibration is off, then adjustments are warranted (again the QA standards should signal this), but if QA standards fall within a predetermined accuracy, then a "recovery" correction is questionable.

Response: We agree that the "recoveries" are not used in the traditional CRM sense.

As the reference solutions were freshly made from new CRM standard solutions, we think "Detection accuracy" may be a more suitable term than "Recovery". Hence, the term "Recovery" has been replaced by "Detection accuracy (DA)" in the manuscript. Also, as each lab didn't use CRM for recoveries, we used the "Detection accuracy (average of 3 detections)" for correction to showcase the difference between corrected and uncorrected values, and also to emphasize the need of using CRM for calibration check and quality control.

Each of the labs prepared the calibration standards by themselves from single solid/liquid standard or dilution of certified standard solutions. CRM wasn't used for recoveries in each lab. We prepared fresh reference solutions and marked them as unknown samples for each lab to analyse. Each lab was asked to follow their own extraction and detection procedures. They were also asked to provide the results of 3 water blanks before the analysis. The calibration and possible QA/QC details of 10 labs have been provided in Table S2. It is possible that some labs did not yet implement strict QA/QC measures strictly as a routine procedure.

In our results, the coefficient of variation (CV) of  $SO_4^{2-}$  and  $NH_4^+$  increased after correction, but it decreased when excluding those labs with high values of DA (>110%). This suggest the need to use CRM for calibration check and quality control. Calibration standards probably should be re-prepared when the DA is large than 110%. This could potentially provide some guidance for future IC analysis. A recommendation has been added to the revised manuscript based on the discussion above. Please see **line 612-615** in the revised manuscript: "Certified reference materials should be used on a regular basis to assess the accuracy and reliability of the measurement method. Calibration standards should be re-prepared and the IC performance should be checked when the detection accuracy is largely deviated from 100% (e.g., > 110% or < 90%)."

Meanwhile, we recognize that such single point correction could be biased. Therefore, we have modified the original sentence "The recovery of the certified reference materials was used to correct the ion concentrations in this study." to "The detection accuracy of the certified reference materials was used to correct the ion concentrations in this study to show the importance of using CRM for calibration check and quality control." in the revised manuscript line 386-388.

The sentence "These results suggested that certified reference materials can be used to correct the Cl- concentrations for more accurate results, especially for less polluted samples." in original manuscript line 392-393 has been deleted.

The sentence "To sum up, certified reference materials should be applied for the correction of the ion concentrations. But the extreme recoveries with large inter-CRM variations should be avoided from the corrections, as this may increase the uncertainty of measurements." has been revised as "To sum up, certified reference materials should be applied for the quality control. If the values of DA are highly deviated from 100% (e.g., >110% or <90%) or there is large inter-CRM variations, then the measurement procedures have to be checked, including repeating the analysis or re-preparing the calibration standard solutions." in the revised manuscript line 420-423.

The sentence "Actual aerosol observations should be corrected for CRM recoveries. But the recovery of ions with poor repeatability should be not be used for correction, as it will cause a larger discrepancy." in original manuscript line 578-580 has been deleted.

**Comments 3:**

While not the central focus of the manuscript, the differences between the ICs and the ACSMs are significant and deserve more attention. The explanation in lines 312-314 seems highly unlikely (volatilization from filters) since sulfate had very similar behavior. Two ideas that were not discussed but which could have contributed to the discrepancies are: 1) differences in the performance of the two PM2.5 cut-point selectors, which could lead to different transmission of particles in the 2-3 µm range, and 2) the collection efficiency applied for the ACSMs. On the second point, the collection efficiency is often assumed to be 0.5, and is applied to an entire data set even though this changes with aerosol composition and meteorological conditions. Therefore, an applied CE of 0.5 when the actual CE was closer to 1 would also produce the observed results. Further, line 339-340 states that "it is essential that the filter-based observations are robustly quality controlled before any ACSM and IC intercomparison," but what about robust quality control of the ACSM measurements? What was done for this study, and what improvements should be made?

**Response:** We agree with the reviewer that the differences in the performance of the two PM2.5 cut-point selectors could lead to different transmission of particles. Regarding the second point, the collection efficiencies (CE) applied for the ACSM at IAP and BUCT were different. For IAP, a capture vaporizer was used, and CE close to 1 was considered as robust (Sun et al., 2020). For BUCT, a standard vaporizer was used. The CE is hence composition- and acidity-dependent and was calculated according to Middlebrook et al. (2012). Therefore, uncertainties of CE may have contributed to the differences between the ACSM and IC results. The CE values used for IAP and BUCT have been added to the revised manuscript in **line 140-143**: "The collection efficiencies (CE) applied for the ACSM at IAP and BUCT were different. For IAP, a capture

vaporizer was used, and the CE was assumed to be close to 1 (Sun et al., 2020). For BUCT, a standard vaporizer was applied with a composition- and acidity-dependent CE calculated according to Middlebrook et al. (2012).".

We have modified the original sentence "A potential reason is the high volatility of these species which leads to higher concentrations in the online ACSM observations compared to the daily filter sample measurements due to negative filter artefacts." as follows:

"Higher concentrations in the online ACSM observations compared to the daily filter sample measurements may be partially due to differences in the performance of the two  $PM_{2.5}$  cut-point selectors, which lead to different transmission efficiency of particles. Other reasons could be: 1) the uncertainties in ACSM observations themselves. Crenn et al. (2015) reported the uncertainties of  $NO_3^-$ ,  $SO_4^{2-}$ , and  $NH_4^+$  in ACSM analysis were 15%, 28%, and 36%, respectively; 2) negative filter artefacts, such as volatilization of semi-volatile ions (Kim et al., 2015), although that the latter would not be expected to affect sulfate. Sun et al. (2020) also compared ACSM and filter based IC results and showed that the concentrations of  $NO_3^-$ ,  $NH_4^+$  and  $SO_4^{2-}$  in the ACSM measurement were also higher than those of filter-based, although the slopes were smaller than in our study." Please see the modified sentences in line 334-343 in the revised manuscript.

The quality control of ACSM measurements has been discussion elsewhere. Please see added sentence in line 144-145 in the revised manuscript "Details regarding quality control of the ACSM at IAP and BUCT can be found elsewhere (Sun et al., 2020; Liu et al., 2020).".

We agree with the reviewer that both measurements should be robustly quality controlled before intercomparison. Hence, we have modified the sentence as "We emphasize that it is essential that both ACSM and filter-based observations are robustly quality controlled before any ACSM and IC intercomparison." in line 358-359 in the revised manuscript.

**Added references:**

Crenn, V., Sciare, J., Croteau, P. L., Verlhac, S., Fröhlich, R., Belis, C. A., Aas, W., Äijälä, M., Alastuey, A., Artiñano, B., Baisnée, D., Bonnaire, N., Bressi, M., Canagaratna, M., Canonaco, F., Carbone, C., Cavalli, F., Coz, E., Cubison, M. J., Esser-Gietl, J. K., Green, D. C., Gros, V., Heikkinen, L., Herrmann, H., Lunder, C., Minguillón, M. C., Močnik, G., O'Dowd, C. D., Ovadnevaite, J., Petit, J. E., Petralia, E., Poulain, L., Priestman, M., Riffault, V., Ripoll, A., Sarda-Estève, R., Slowik, J. G., Setyan, A., Wiedensohler, A., Baltensperger, U., Prévôt, A. S. H., Jayne, J. T., and Favez, O.: ACTRIS ACSM intercomparison – Part 1: Reproducibility of concentration and fragment results from 13 individual Quadrupole Aerosol Chemical Speciation Monitors (Q-ACSM) and consistency with co-located instruments, Atmos. Meas. Tech., 8, 5063-5087, 10.5194/amt-8-5063-2015, 2015. Kim, C. H., Choi, Y., and Ghim, Y. S.: Characterization of Volatilization of Filter-Sampled PM2.5 Semi-Volatile Inorganic Ions Using a Backup Filter and Denuders, Aerosol Air Qual. Res., 15, 814-820, 10.4209/aaqr.2014.09.0213, 2015

Liu, Y., Yan, C., Feng, Z., Zheng, F., Fan, X., Zhang, Y., Li, C., Zhou, Y., Lin, Z., Guo, Y., Zhang, Y., Ma, L., Zhou, W., Liu, Z., Wei, Z., Dada, L., Dallenbach, K. R., Kontkanen, J., Cai, R., Chan, T., Chu, B., Du, W., Yao, L., Wang, Y., Cai, J., Kangasluoma, J., Kokkonen, T., Kujansuu, J., Rusanen, A., Deng, C., Fu, Y., Yin, R., Li, X., Lu, Y., Liu, Y., Lian, C., Yang, D., Wang, W., Ge, M., Wang, Y., Worsnop , D., Junninen, H., He, H., Kerminen, V. M., Zheng, J., Wang, L., Jiang, J., Petäjä, T., Bianchi, F., and Kulmala, M.: Continuous and Comprehensive Atmospheric Observations in Beijing: A Station to Understand the Complex Urban Atmospheric Environment, Big Earth Data (under review), 10.1080/20964471.2020.1798707, 2020.

Middlebrook, A. M., Bahreini, R., Jimenez, J. L., and Canagaratna, M. R.: Evaluation of Composition-Dependent Collection Efficiencies for the Aerodyne Aerosol Mass Spectrometer using Field Data, Aerosol Sci. Technol., 46, 258-271, 10.1080/02786826.2011.620041, 2012.

Sun, Y., He, Y., Kuang, Y., Xu, W., Song, S., Ma, N., Tao, J., Cheng, P., Wu, C., Su, H., Cheng, Y., Xie, C., Chen, C., Lei, L., Qiu, Y., Fu, P., Croteau, P., and Worsnop, D. R.: Chemical Differences Between PM1 and PM2.5 in Highly Polluted Environment and Implications in Air Pollution Studies, Geophysical Research Letters, 47, e2019GL086288, 10.1029/2019gl086288, 2020.

**Technical Corrections:**

- Line 48: suggest "factor of three" instead of "3 times"

**Response:** Thanks for the reviewer's advice. The original sentence "Cl- from the two methods are correlated but the concentration differ by more than three times." has been changed to "Cl- from the two methods are correlated but the concentration differ by more than a factor of three." Please see line 48 in the revised manuscript.

- Line 69: suggest deleting the second 'to' in this sentence

Response: This has been revised

- Line 77-78: sentence awkward as written, suggest rephrasing

**Response:** This has been rephrased as "However, previous methods were timeconsuming as WSII were analyzed by different techniques separately.". Please see line 78-79 in the revised manuscript.

- Line 84: suggest replacing "nowadays" with "at present" (or similar) **Response:** This has been revised

- Line 183: change to "COD value equal to..." **Response:** This has been revised

- The COD value of 0.269 (line 187-189, Figure 4) seems highly arbitrary. There really is no discussion about the 0.20 vs. 0.269 - I suggest picking only one.

**Response:** Thanks for the reviewer's suggestion.

The COD value of 0.269 has been deleted from the original Figure 4 (revised Fig. 3) and Fig. S1-S3, as well as in the context.

- Line 320: was an statistical outlier test actually performed? Be careful excluding data without reason, especially for a smaller n like this study.

**Response:** Thanks for the reviewer's comment. As outlier test was not performed and the number of samples was small. Hence, the data point was not excluded for discussion. We have deleted the original sentence "but  $R^2$  increased to 0.82 when excluding an outlier of the data on 23rd."

- Line 328: suggest "factor of three" instead of "3 times"

**Response:** Thanks for the reviewer's suggestion. "3 times." has been changed to "a factor of three.". Please see line 348 in the revised manuscript.

- Suggest adding 1:1 lines to Figure 2

**Response:** Thanks for the reviewer's suggestion. 1:1 lines have been added to Figure 2. Please see modified Fig. 2 (line 361) in the revised manuscript.

- Section 3.2.3 is quite short, and could be omitted, or incorporated into another section. Fig. 3 does not add much of substance to the overall discussion.

**Response:** Thanks for the reviewer's suggestion. This section has been incorporated into section 3.2.1. The original Fig. 3 has been moved to supplemental information Fig. S1. Please see line 266-270 in the revised manuscript.

- Line 523-524: what is the hypothesized reason for the highly different pH value from Lab-9's data on this date?

**Response:**

We have run some sensitivity tests to investigate the hypothesized reason for the abnormal pH value of Lab-9 on  $20^{th}$  January. There is no significant difference when we change the input concentrations of K+, Mg2+ and Ca2+. However, when we change the original concentration of Na+ (0.19 µg/m3) to the median value of 10 labs (0.03 µg/m3) on that day, the pH increased from 5.8 to 7.5, which is similar to those of other labs. Hence, we think it was because of the abnormal Na+ concentration on that day of

Lab-9 (Fig. 1) that caused the highly different pH value. Please see our modified text below (line 553-554 in the revised manuscript):

"Sensitivity test of Na+, K+, Mg2+ and Ca2+ showed that this abnormal pH value was mainly due to the significant higher Na+ concentration of Lab-9 on  $20^{\text{th}}$ ."

- Line 565: I disagree that there are "substantial" uncertainties, especially in the major ions measured by IC. The uncertainties are quantifiable, and they seem to be in line with previously published values. For the "minor" ions, the threshold for major/minor is never stated, but clearly this is true for Ca2+, while most of the others seemed to perform quite well.

**Response:** Thanks for the reviewer's suggestion. We have modified the original sentence "There are substantial inter-lab uncertainties in both the aerosol major and minor ions measured by ion chromatography from the filters." to " The uncertainties are particularly large for minor ions like  $Ca^{2+}$  from the aerosol filters-based ion chromatography analysis." In line 606-607 in the revised manuscript.

- Line 582: yes, but I would note here that the median ammonium recovery was close to 100%.

**Response:** Thanks for the reviewer's suggestion. The median ammonium recovery was close to 100% is added: "The detection accuracy of ammonium varied significantly among 10 labs (88.4-135.0%) with median value close to 100%.". Please see line 616-617 in the revised manuscript.

---

## Author Response (AR2)

**Point-by-Point Response to Reviewer #2's further Comments**

Manuscript Ref: amt-2020-156

Title: An inter-laboratory comparison of aerosol in organic ion measurements by Ion Chromatography: implications for aerosol pH estimate

Journal: Atmospheric Measurement Techniques

**Further comments from Reviewer #2**

**General Comments:**

I still have one concern about the influence of NH3 on the estimation of aerosol pH, as follows:

1. The NH3 concentration in this study was so high that always enough to neutralize the acidic components. Even assuming the uncertainty of ± 10 ppb, the low limit of NH3 concentration would be larger than 5 ppb, which was still high enough to "constrain" the aerosol pH and result in similar pH obtained among different labs. But it is not necessarily to mean the measurement bias of water-soluble ions is not important.

**Response:** Thanks for the reviewer's valuable comment. To address the concern mentioned by the reviewer, we conducted a sensitivity test for $NH_3$. Please see added discussions below:

"To investigate the effect of $NH_3$ concentration on aerosol pH, we conducted a sensitivity test which showed the aerosol pH of samples measured by 10 labs at $NH_3$ levels of 0.5, 1, 2, 5 and 10 ppb (Fig. S8). When the concentration of $NH_3 \geq 2$ ppb, the aerosol pH estimates of the 10 labs were generally consistent and less affected by the variation of ion concentrations. But there is more variation of aerosol pH in the 10 labs when $NH_3$ concentration was under 2 ppb. This suggests when $NH_3$ concentration $< 2$ ppb, the aerosol pH could be more affected by the variation of ion concentrations."

Please see **line 568-574** in the revised manuscript.

2. The NH3 measurement in this study was a daily averaged value. BUT NH3 would have strong diurnal variation. Although the average concentration was very high, it was still possible that NH3 concentration was low enough to induce a large change of aerosol pH.

**Response:** Thanks for the reviewer's comment. We recognize that there is a diurnal variation in aerosol composition and $NH_3$ concentrations. However, as mentioned in the manuscript, the daily ammonia concentrations during the study period ranged from 13.9±0.6 to 20.1±0.7 ppb. The small standard deviation (<1ppb) of the daily average (derived from the original 5-minute data during each sampling day) suggest the diurnal variation of $NH_3$ concentration was not significant and would not cause a large change

of aerosol pH, especially when the $NH_3$ concentrations were relatively high. In addition, the lowest 5-minute average $NH_3$ concentration during the whole sampling period was 12.4 ppb, higher than 10ppb. Fig. S8 shows that the results of aerosol pH in 10 labs at 10ppb were consistent. Hence, we believe the diurnal variation of $NH_3$ in this study would not induce a large change of aerosol pH. However, future studies should consider the impact of the diurnal variation of $NH_3$ on aerosol pH if the $NH_3$ concentration was lower than 2 ppb, as mentioned in point 1 above.

The sentence " The daily ammonia concentrations during the study period ranged from 13.9±0.6 to 20.1±0.7 ppb with an average of 17.2±2.2 ppb." has been changed as

"The daily ammonia concentrations during the study period derived from 5-minute data ranged from 13.9±0.6 to 20.1±0.7 ppb (average: 17.2±2.2 ppb). The small standard deviations of the daily average (< 1 ppb) suggest that the diurnal variation of $NH_3$ was not significant. Hence, aerosol pH was only investigated using daily mean $NH_3$ concentrations." Please see **line 515-518** in the revised manuscript.

A recommendation is added based on the above discussions.

"The variation of ion concentrations is expected to strongly affect aerosol acidity estimated by ISORROPIA II when the $NH_3$ concentration is low (e.g., < 2 ppb in this case). Additionally, the impact of the diurnal variation of $NH_3$ on aerosol acidity is worthy of investigation, particularly when the $NH_3$ concentration is low." Please see **line 620-623** in the revised manuscript.